# Stress-induced ordering evolution of 1D segmented heteronanostructures and their chemical post-transformations

Qing-Xia Chen[1,3], Yu-Yang Lu[2,3], Yang Yang[2,3], Li-Ge Chang[2], Yi Li[1], Yuan Yang[1], Zhen He[1], Jian-Wei Liu[1]✉, Yong Ni[2]✉ & Shu-Hong Yu[1]✉

Investigations of one-dimensional segmented heteronanostructures (1D-SHs) have recently attracted much attention due to their potentials for applications resulting from their structure and synergistic effects between compositions and interfaces. Unfortunately, developing a simple, versatile and controlled synthetic method to fabricate 1D-SHs is still a challenge. Here we demonstrate a stress-induced axial ordering mechanism to describe the synthesis of 1D-SHs by a general under-stoichiometric reaction strategy. Using the continuum phase-field simulations, we elaborate a three-stage evolution process of the regular segment alternations. This strategy, accompanied by easy chemical post-transformations, enables to synthesize 25 1D-SHs, including 17 nanowire-nanowire and 8 nanowire-nanotube nanostructures with 13 elements (Ag, Te, Cu, Pt, Pb, Cd, Sb, Se, Bi, Rh, Ir, Ru, Zn) involved. This ordering evolution-driven synthesis will help to investigate the ordering reconstruction and potential applications of 1D-SHs.

Regular pattern design in one-dimensional segmented hetero-nanostructures (1D-SHs) are fundamentally crucial due to the synergistic effect between different components and interfaces[1–5]. Controlled position of various materials within one nanostructure is significantly important for the function integration, such as steering a tandem reaction[6]. Heterogeneous interfaces determine the electronic and magnetic coupling between multiple compositions, which can facilitate the electron/hole transport in photocatalytic water splitting and boost the phonon scattering in thermoelectric application[7–10]. Therefore, the rational design and precise synthesis of 1D-SHs plays a decisive role in the applications of next-generation nanostructures across many fields. Currently, several methods have been developed to fabricate 1D-SHs, such as vapor-liquid-solid growth[11–15], solution-liquid-solid growth[16,17], cation exchange[18–21], confined template precipitation[22,23], and epitaxial growth[24,25]. However, additional control over the division compositions, segment separations, and interface types, is still a formidable hurdle.

Meanwhile, these proposed reaction mechanisms of precursor-sequenced addition[26], diffusion-limited ordering[27], and strain field induction[28] are worth more effort to further improve the growth mechanism at nanoscale[26].

In addition, only limited heterostructured 1D nanomaterials, such as sulfides and selenides, have been prepared[29–32]. Telluride materials offer significant advantages in terms of their high theoretical volume capacity and high electrical conductivity, making them promising candidates for different energy storage and optoelectronic applications[33,34]. Meanwhile, the higher density of tellurides compared to sulfides and selenides produces a higher volume energy density, which also holds great promise for the battery field. For example, transition metal telluride, like other layered materials, possess a distinctive nanostructure with large interlayer spacing, which is conducive to the rapid transfer of ions in the electrode[35]. These benefits make telluride-based nanomaterials an attractive option for various

[1]Department of Chemistry, Institute of Biomimetic Materials & Chemistry, Anhui Engineering Laboratory of Biomimetic Materials, Division of Nanomaterials & Chemistry, Hefei National Laboratory for Physical Sciences at the Microscale, University of Science and Technology of China, Hefei 230026, China. [2]CAS Key Laboratory of Mechanical Behavior and Design of Materials, Department of Modern Mechanics, University of Science and Technology of China, Hefei 230026, China. [3]These authors contributed equally: Qing-Xia Chen, Yu-Yang Lu, Yang Yang. ✉e-mail: jwliu13@ustc.edu.cn; yni@ustc.edu.cn; shyu@ustc.edu.cn

energy-related applications. Therefore, a simple and scalable platform for constructing the 1D telluride SHs library, explaining their whole evolution mechanism, as well as exploring their promising applications, has become a crucial issue to resolve[36].

The continuum phase-field model has recently emerged as a simulation method to model and predict the mesoscale morphology and microstructure evolution in materials, especially working in describing the phase transformation coupled with multi-field physics[37,38]. A set of conserved and non-conserved field variables across the smoothed interface, including the thermodynamic, kinetic and mechanic information, are used to simulate the temporal and spatial distributions of concentration and stresses[39,40]. Herein, we exploit a simple and versatile approach towards solution-synthesized 1D telluride SHs based on the under-stoichiometric reaction strategy. Using this strategy and easy chemical post-chemical transformations, we synthesized about 25 periodic 1D-SHs, including 17 nanowire-nanowire (NW-NW) and 8 nanowire-nanotube (NW-NT) nanostructures with 13 elements involved. Specifically, we established the stress-induced ordering mechanism by phase-field model to describe the rationalization of ordering evolution from poor-ordered stripes into periodic segments. And the three-stage evolution process, *i.e.*, island generation, stripe penetration and segment ordering, is further proposed. To demonstrate the superiority of segmented heterostructures, $Ag_2Te/PbTe$ SHs are selected as prototype for thermoelectric performance testing. The current methodology may affect the fabrication of function-oriented nanomaterials as well as the comprehension of periodic ordering in 1D intricate nanostructures.

## Results

### Te/Ag₂Te 1D SHs and their segment evolution

We selected Te NWs with a high aspect ratio[41] as the model structural unit. After adding complexing agent $NH_4SCN$ and under-stoichiometric $Ag^+$ in Te NW solution, the reaction completed in a few minutes accompanied by the color changing from mazarine to brown. The fabricated $Te/Ag_2Te$ NWs show the segmented heterogeneous nanostructure (Fig. 1a). The large-scale TEM image of $Te/Ag_2Te$ NWs is displayed in Suppl. Figure 1, which indicates that the segmented structure is uniform. As observed from the EDS composition mappings in Fig. 1b, $Te/Ag_2Te$ NWs are featured by axially even Te and discrete Ag distributions. The HRTEM image in Fig. 1c highlights the interface of a single NW. Lattice spacings of 0.194 and 0.201 nm are indexed to ($\bar{4}02$) plane of Te and (100) plane of $Ag_2Te$, respectively. A view of the atomic interface in Fig. 1d illustrates that Te and $Ag_2Te$ form their lattices independently, showing that $\{\bar{2}04\}$ facet of $Ag_2Te$ binds to {001} facet of Te. The biphasic attribute is further vindicated by XRD pattern in Fig. 1e, in which all peaks are assigned to either hexagonal Te (PDF#36-1452) or monoclinic $Ag_2Te$ (PDF#34-0142) (Suppl. Figure 2e). In addition to XRD pattern, the Raman spectra of Te, $Ag_2Te$, and $Te/Ag_2Te$ NWs are also captured. As shown in Suppl. Figure 2f, the peaks located at 125 and 2915 cm⁻¹ are related to $A_1$ mode of Te, corresponding to the chain expansion mode[42]. The peak located at 650 cm⁻¹ is indexed to $Ag_2Te$, due to its decomposition under laser-beam irradiation[43]. Moreover, XPS spectra of Te 3d orbital of the SHs in comparison with Te and $Ag_2Te$ NWs are shown in Fig. 1f. The binding energy of Te $3d_{5/2}$ in $Te/Ag_2Te$ located at 575.95 eV, while it is 575.84

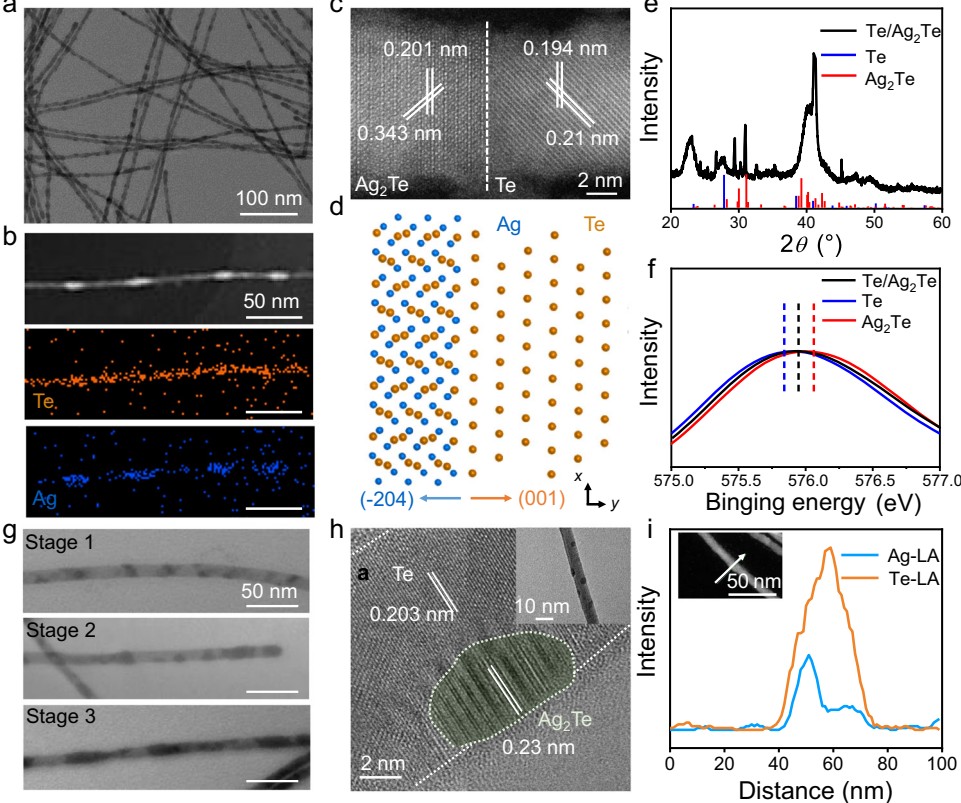

**Fig. 1 | Morphological and structural characterizations of Te/Ag₂Te SHs.**
**a** Typical TEM image, revealing the segmented periodic heteronanostructure.
**b** HAADF and EDS mapping, showing the homogeneous distribution of Te (orange) and heterogeneous distribution of Ag (blue). All scale bars are 50 nm. **c** HRTEM image, showing the specific interface between Te and Ag₂Te. **d** A schematic atomic model, specifying an atomically sharp interface in heteronanostructures. The hexagonal Te (001) plane connects with the monoclinic Ag₂Te (−204) plane. **e** XRD pattern, showing the coexist of Te and Ag₂Te. **f**, XPS spectra of Te 3d orbital of the heteronanostructures. **g** Time-resolved TEM images. t = 10 s, 30 s and 3 min for Stage 1, 2, and 3, respectively. t is the reaction time between Te and Ag⁺. **h**, HRTEM image of the island on the NW. Inset: TEM image of the nanostructure formed at the beginning. **i** The distribution of Ag₂Te island formed on the Te NW, obtained by line analysis of EDS. Inset: HAADF image of island formed on the NW. Source data are provided as a Source Data file.

and 576.06 eV for Te and $Ag_2Te$, respectively. This shift is due to the presence of both $Te^0$ and $Te^{2-}$ subphases, which corroborates well with previous observation[44]. The peak deconvolution results show that $Te^{4+}$ inevitably exists due to the oxidization of Te in the presence of air (Suppl. Figure 2h). The detailed morphological and structural characterizations of Te/$Ag_2Te$ NWs compared with pure Te and $Ag_2Te$ NWs are included in Suppl. Figure 2. The Te and $Ag_2Te$ segment control are obtained with Ag feeding variations shown in Suppl. Figure 3 and Suppl. Table 1.

To probe into the growth processes of Te/$Ag_2Te$ SHs, we captured their morphologies at different times shown in Fig. 1g and Suppl. Figs. 4, 5. Due to the high reactivity of Te NWs with $Ag^+$, $NH_4SCN$ was used as a complexing agent to dramatically decelerate the reaction. During the very early stage, Te NW is scattered randomly with few islands (Suppl. Figs. 4a, 5a). As the reaction proceeds, the island embryo initiates to propagate radially throughout the diameter, generating the well-defined strips (Suppl. Figs. 4b, 5b). With the prolonged reaction time, these disordered strips reconstruct into ordered heteronanostructures (Suppl. Figs. 4c, 5d). To visualize the initial reaction period, ice bath was employed to further reduce the reaction rate. As shown in Suppl. Fig. 6, the initial $Ag_2Te$ island is embedded into Te NW templates with a very small size. The lattice distances of 0.203 and 0.23 nm in Fig. 1h match that of Te and $Ag_2Te$, respectively. The HADDF images in Suppl. Fig. 7 indicates that the island insets into Te NW with a triangle front. The element distributions in Suppl. Fig. 7 displays that the islands are distributed with Te and Ag. The intensity profile in a line scan across the island on NW in Fig. 1i further demonstrates that Te is evenly distributed on the NW, but Ag is almost dispersed on the island. It is worth noting that the signal strength of Te is about twice of Ag, which proves that the islands are $Ag_2Te$ rather than Ag. This is also consistent with the HRTEM image of island, which shows the lattice fringe of $Ag_2Te$. As the reaction progressed, the islands grew larger and more numerous. Then these islands continued to develop, and some islands begun to permeate throughout the diameter, forming the stripped structures. Additionally, the spectra changes in the evolution of Te/$Ag_2Te$ SHs are also shown to describe the average change in the morphology and structure. As the UV-vis spectrum in Suppl. Fig. 8a shows, when $Ag^+$ is added to Te NW solution, two shoulder peaks appear at 464 and 566 nm expect the absorption peaks of Te NWs. These two absorption peaks result from the formation of heterogeneous interfaces between Te and $Ag_2Te$. This can be further confirmed by the fluorescent emission spectra in Suppl. Fig. 8c, where the peak at 454 nm of $Ag_2Te$ gradually increases with the evolution from island to segmented structures.

To further delve into the initial structural evolution, in-situ environmental liquid-state cell in TEM was used to study the real-time forming and growth of island. As $Ag^+$ was just injected, Te NW roughened immediately, and obvious humps appeared (Suppl. Fig. 9 and Suppl. Movie 1). These humps continued to grow into well-defined islands with the emergence of small bulges. The reaction between Te NW and $Ag^+$ was so fast that visible bulge formation could be observed within 1 s. These islands were embedded into the body of NW through the entire diameter of NW. The bulge penetrated the diameter within 5 s. This island forming and growth process were also confirmed by the change of island forming area ($C_i$) and forming rate ($R_i$). As shown in Suppl. Fig. 9, $C_i$ increased as a function of time, showing the bulges keeping growing. $R_i$ increased rapidly due to the more and more island generating sites.

The continuum phase-field model was employed to reveal the mechanism of the observed kinetic process via simulating the evolution of Ag concentration in Te NW. The intact Te NW with the surrounding solution in the computational model is shown in Fig. 2a. The morphology of Te NW is characterized by a domain parameter $\varphi$, which equals 0 in solution and 1 in NW, respectively. By introducing another order parameter $c$, the Ag induced phase transformation in

NW can be described (Suppl. Table 2). When $\varphi = 1$ and $c = 1$ indicates the $Ag_2Te$ phase, while $\varphi = 1$ and $c = 0$ indicates the intact Te NW phase. The kinetic processes of $Ag_2Te$ island formation, coarsening into strips and transition into equidistant patterns are realized (Fig. 2b and Suppl. Movie 2). We assume that the surface defect is heterogeneously distributed on the surface of Te NW. $Ag^+$ in solution is preferentially absorbed to the site with high defect density on Te NW surface and reacts with Te due to the lower reaction energy barrier. Once the $Ag_2Te$ island embryo forms, it will rapidly develop towards the radial direction and form the strip pattern, rather than grow along the Te NW surface to form core-shell pattern. It should be pointed out that metal precursors ($Ag^+$, $Cu^{2+}$, $Bi^{3+}$, $Pb^{2+}$ and $Cd^{2+}$) cannot be reduced by Te according to the standard reduction potential sequence[34] listed in Fig. 2c. In this case, homogeneous alloy, e.g., AgTe NW, cannot be obtained. Figure 2d shows the schematic diagram of a generated $Ag_2Te$ island, with purple parallelogram indicating the interface between Te and $Ag_2Te$. By analyzing the stress distribution at the interface, we demonstrate that the stress caused by the embedded Ag in the front of island is tensile, which promotes radial growth. Figure 2e exhibits the dimensionless Ag concentration and stress distribution at the interface, respectively. The front of the island is tensile hydrostatic stress while the sides are compressive hydrostatic stress, which promotes the further radial growth for $Ag_2Te$ island. Note that the guest species favors migrating into the tensile hydrostatic stress region in the host material to decrease the chemical potential[39,40]. This explains that the radial growth of island is faster than the axial growth in experiment, prohibiting the formation of core-shell nanostructures. Moreover, we discuss the effect of the anisotropic interfacial energy on the growth process of the $Ag_2Te$ island (Suppl. Figs. 10, 11). In the Te/$Ag_2Te$ system, it is demonstrated that the stress plays a much more important role than interfacial energy, although interfacial energy has a limited influence on the growing dynamics. In addition, by modifying the phase field model, how the interfacial and surface diffusion affect the $Ag_2Te$ islands growth is investigated separately (Suppl. Fig. 12). We find that the fast interfacial diffusion process promotes the radial growth of $Ag_2Te$ islands, thereby accelerating the strip pattern formation in the NW, while the fast surface diffusion favors to the longitudinal growth of $Ag_2Te$ islands, which is not conducive to the strip pattern formation. Furthermore, the strip formation is confirmed by analyzing Ag concentration development in a cylindrical zone containing the $Ag_2Te$ island shown in Fig. 2f. It indicates that the average dimensionless concentration $\left(c_{equal}/c_{max}\right)$ of the selected zone increases and then saturates with time from island generation to strip penetration in Fig. 2g. To summarize, we illustrate the three-stage evolution from Te NWs into Te/$Ag_2Te$ SHs in Fig. 2h, ie., defect-assisted island generation, hydrostatic tensile stress-favorable radial growth, and elastic energy-driven ordering. The ordering process will be discussed specifically below.

## Ordering of Te/$Ag_2Te$ 1D SHs

To uncover the underlying mechanism of the ordering of Te/$Ag_2Te$ SHs, the transition from poorly ordered striped nanostructure to well-ordered segmented heteronanostructure is systematically investigated. Through theoretical simulation of the ordering shown in Fig. 3a, the segments close to each other attract and merge into a continuum, while the segments far away repel each other, finally forming the segment Te/$Ag_2Te$. The transition from a poorly ordered structure to well-ordered pattern is also confirmed by statistically analyzing dimensionless lengths of Te ($H/a$) and $Ag_2Te$ ($h/a$) segments in Fig. 3b. $a$ is the radius of NW. $H$ and $h$ are the lengths of segmented Te and $Ag_2Te$, respectively. From the histograms, both $Ag_2Te$ and Te segments transform from a very wide distribution in poorly ordered SHs to a centralized distribution in well-ordered SHs, which demonstrates the equidistant pattern formation of both $Ag_2Te$ and Te segments.

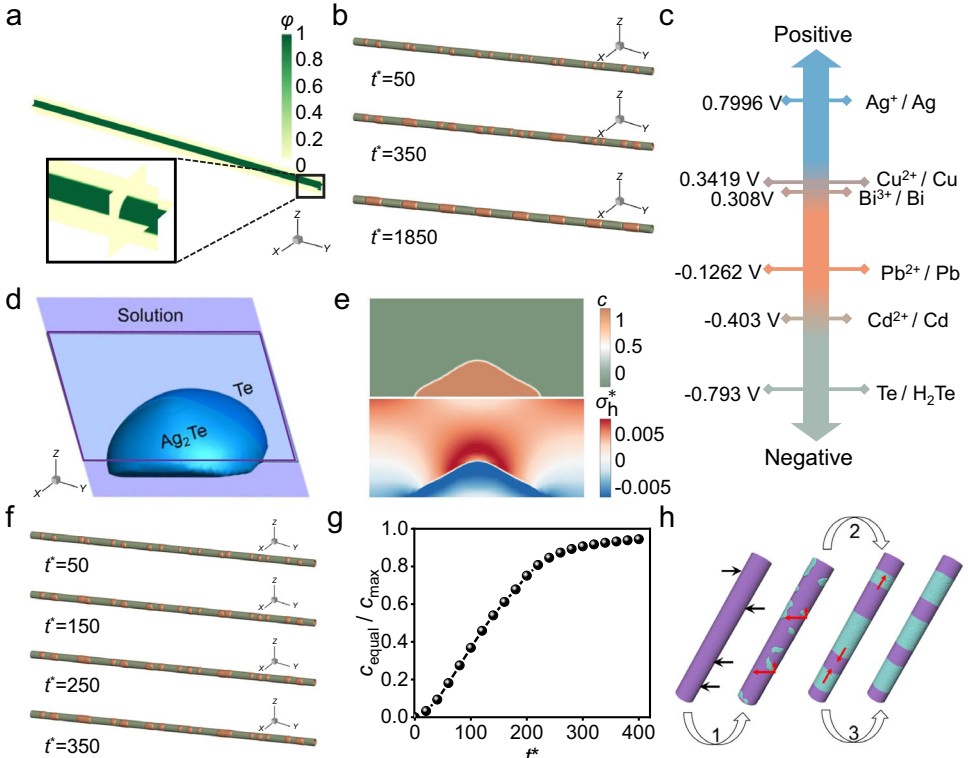

**Fig. 2 | Exploration on the formation process of Te/Ag₂Te SHs. a** The NW (green color) and the surrounding solution (yellow color) in the computational model. The morphology of the Te NW is characterized by a domain parameter $\varphi$, which equals 0 in the solution and 1 in the nanowire, respectively. **b** Time-resolved simulation images. $t^*$=50, 350, and 1850 for Stage 1, 2 and 3, respectively. $t^*$ is the simulated dimensionless reaction time between Te and Ag⁺. **c** Reduction potentials of Te and various metal in acidic medium. **d** Schematic diagram of the generated Ag₂Te island. The purple parallelogram represents the interface between Te and Ag₂Te.

**e** Sectional views of the dimensionless concentration and average stress distributions in the generated Ag₂Te island. **f** Dimensionless concentration evolution results of the poorly ordered SHs at times of $t^*$= 50, $t^*$= 150, $t^*$= 250 and $t^*$= 350. **g** Simulated concentration evolution of Ag in the selected zone near the starting point of flow. **h**, Schematic illustration of three stages involved in the evolution of the periodic heteronanostructure. The $\varphi$, concentration, and average stress in this model have been dimensionless processed to perform the numerical simulations. Source data are provided as a Source Data file.

By magnifying a representative zone, Fig. 3c illustrates the kinetic process of Te/Ag₂Te SH formation. In this schematic diagram, two segments of Ag₂Te (brown), one segment of Te (olive) and the surrounding Ag⁺ solution (blue) are selected for study. The gray balls refer to the free-migrating Ag⁺. The white balls are Ag⁺ that has migrated to Te segments, while the red ones are the Ag⁺ moved back to Ag₂Te segments due to the high mobility and equilibrium. The black arrows refer to the migration of Ag⁺ between the NW solid and surrounding solution. There is no Ag in Te phase in the ideal Te/Ag₂Te nanostructure after the generation of poorly ordered stripes. During the relaxation, infinitesimal Ag in Ag₂Te segments diffuses into the neighboring clean Te due to the high mobility of Ag[38]. It should be noted that the migrated Ag cannot localize in Te region and stabilize the interface, which means that little Ag⁺ shuttles back and forth between two phases. Therefore, Te and Ag₂Te segments both experience the ordering. Ag⁺ migration causes the interface between Te and Ag₂Te to move, forming the segmented structure with uniform length distribution.

Moreover, in this ordering stage, we first construct two Ag₂Te segments with a constant spacing $H$ and keep the total length of two Ag₂Te segments constant. When we change the difference between the lengths of two Ag₂Te segments $(h_1 - h_2)$, we can obtain the variation of elastic energy as a function of $(h_1 - h_2)$. Second, we construct one Ag₂Te and two Te segments. Similarly, we set the length Ag₂Te segment $(h)$ to be constant and keep the spacings of Te segments, $(H_1 - H_2)$, unchanged. Then, we alter the length difference of two Te segments $(H_1 - H_2)$, and calculate the elastic energy under different $(H_1 - H_2)$. As shown in Fig. 3d, we explain the formation of the

equidistant strips by analyzing the evolution of elastic energy. The dimensionless elastic energy $\left(W^*\right)$ of the system increases monotonically with increasing the discrepancy of the length between adjacent Ag₂Te $(\Delta h = h_1 - h_2)$ and Te segments $(\Delta H = H_1 - H_2)$, respectively. The elastic energy reaches a minimum value at $(\Delta h = 0)$ and $(\Delta H = 0)$. Figure 3e shows that the elastic energy rapidly climbs up to a maximum, then decreases slowly until finally stabilizes with the increase of $H/a$ (the dimensionless spacing between Ag₂Te). We define the spacing corresponding to the maximum elastic energy as the critical spacing. When the spacing between Ag₂Te segments is less than the critical value, the elastic energy will decrease rapidly as the spacing decreases. Therefore, the adjacent Ag₂Te segments show an attractive interaction and eventually merge into one segment. On the contrary, if the segment spacing is larger than the critical value, the elastic energy will decrease with the increase of spacing. As a result, two Ag₂Te segments exhibit repulsive interaction and are prone to increase the spacing (Suppl. Fig. 13 and Suppl. Movie 3). It should be pointed out that the radius of NW is not an independent factor that affects the elastic driving force for the ordering of two adjacent segments. The ratio of Ag₂Te segment spacing to the radius $(H/a)$ and the ratio of Ag₂Te segment length to the radius $(h/a)$ are the dominated factors that affect the elastic driving force.

Following the same vein, we fabricated Te/PbTe, Te/Cu₁.₇₅Te, Te/CdTe, and Te/Bi₂Te₃ SHs after feeding G.1 Te NW templates with insufficient corresponding metal precursors shown in Suppl. Fig. 14. When adding the under-stoichiometric Pb precursor with Te/Pb=3.2 into Te NWs, Te/PbTe SHs can be obtained[39]. The uniformity is confirmed by the large-scale TEM image in Suppl. Fig. 15. The HADDF-

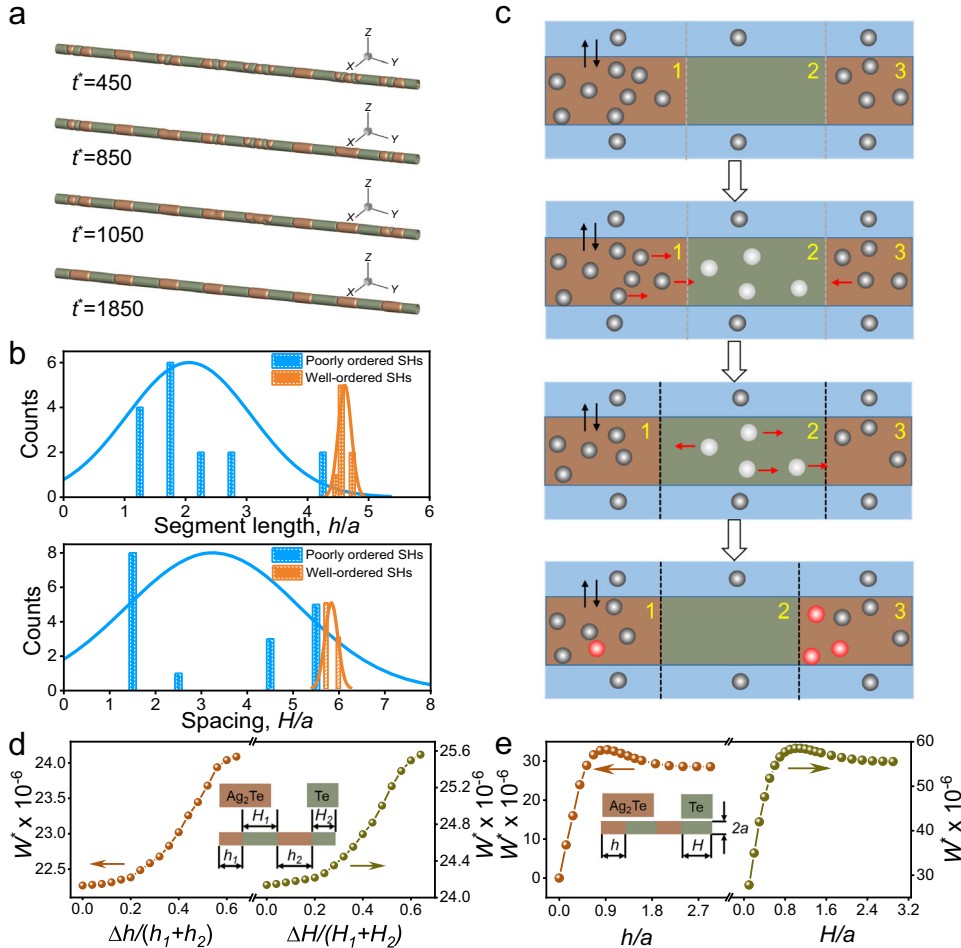

**Fig. 3 | Ordering evolution in Te/Ag$_2$Te SHs. a** Dimensionless concentration evolution from disordered strips to ordered segments in the NW. **b** Statistical distribution comparisons of Ag$_2$Te (upper) and Te (below) segments with various lengths, where poorly and well-ordered SHs correspond to $t^*$ = 450 and $t^*$ = 1850, respectively. The solid line represents the Gaussian distribution curve fitted to the data. **c** Schematic diagram of the interface movement in the segmented NW imposed by the Ag migration. The blue-columns represent the Ag$^+$ solution surrounding the NW, while the brown and olive-columns represent Ag$_2$Te and Te segments, respectively. **d, e** The calculated dimensionless elastic energy versus the differences in length between adjacent segments (**d**) and ratios of segment separation to the radius and segment length to the radius (**e**). Source data are provided as a Source Data file.

STEM and elemental mapping characterizations in Suppl. Fig. 16 shows that Te is distributed homogeneously while Pb heterogeneously along the axial direction. HRTEM images show the spacings of 0.218 and 0.224 nm corresponding to Te and PbTe, respectively. Actually, a series of Te/PbTe SHs with different PbTe and Te segment lengths are simply produced with varying Te/Pb ratios (Suppl. Fig. 17). Particularly, the as-prepared Te/PbTe SHs resemble the shish-kebab-like structure attributed to the large lattice parameter differences between Te and PbTe. The Te/Cu$_{1.75}$Te and Te/CdTe SHs are also identified in Suppl. Figs. 18, 19, respectively. As a note, Bi$_2$Te$_3$ tends to develop into lamellas owning to its anisotropic growth[40] and Te/Bi$_2$Te$_3$ SHs consequently show a rough surface (Suppl. Figs. 20, 21). Bi$_2$Te$_3$ flakelets threaded into Te NW are obviously identified with tiny Bi addition.

Phase-filed model is also adopted to calculate the elastic energy change in the evolution of Te/PbTe and Te/Cu$_{1.75}$Te SHs (Suppl. Figs. 22, 23). Similar with Te/Ag$_2$Te SHs, the elastic energy of Te/PbTe SHs increases monotonously with spacing difference of both PbTe and Te segments increasing. In other words, Te and PbTe segments are both apt to be isometrically distributed despite the larger lattice mismatch in Te/PbTe than Te/Ag$_2$Te. It is worth pointing out that the large mismatch strain facilitates the coalesce of the adjacent segments due to the large driving force (Suppl. Fig. 22). In addition, Suppl. Fig. 23b shows that both PbTe and Te segments experienced

the peak elastic energies and critical values subordinating to the stress-induced ordering mechanism. The energy-induced structural evolution of Te/Cu$_{1.75}$Te SHs also verifies this mechanism (Suppl. Fig. 23c, d).

## Chemical transformations from G.1 and G.2 templates

With elaborate design of the chemical transformations from G.1 Te NWs and G.2 Te/M$_1$Te SHs templates, we fabricated another 20 SHs shown in Fig. 4a. The generated M$_1$Te can be exchanged with other metal cations. Taking Te/Ag$_2$Te as an example, Ag$^+$ is replaced by Zn ion[41]. The derivative Te/ZnTe SHs inherit the segmented structure (Suppl. Fig. 24). Moreover, the redundant Te with high reactivity can be further converted into M$_3$Te. Metal precursor M$^{3+}$ (M$_3$=Cu, Cd, Bi, Sb) can be easily integrated into G.2 templates with accurate feeding ratios, producing Cu$_{1.75}$Te/Ag$_2$Te, CdTe/Ag$_2$Te, Bi$_2$Te$_3$/Ag$_2$Te and Sb$_2$Te$_3$/Ag$_2$Te SHs (Suppl. Figs. 25–28). Similarly, the G.2 template, Te/PbTe SHs, experiences the chemical post-transformations following the same design roadmap. G.3 SHs including Ag$_2$Te/PbTe, Cu$_{1.75}$Te/PbTe, CdTe/PbTe, Bi$_2$Te$_3$/PbTe, and Sb$_2$Te$_3$/PbTe are produced after combining corresponding cations with Te. Satisfactorily, the classic thermoelectric material Ag$_2$Te/PbTe still remains the segmented morphology in large-scale preparation (Suppl. Figs. 29, 30). Other G.3 M$_3$Te/PbTe SHs are detailed in Suppl. Figs. 31–34.

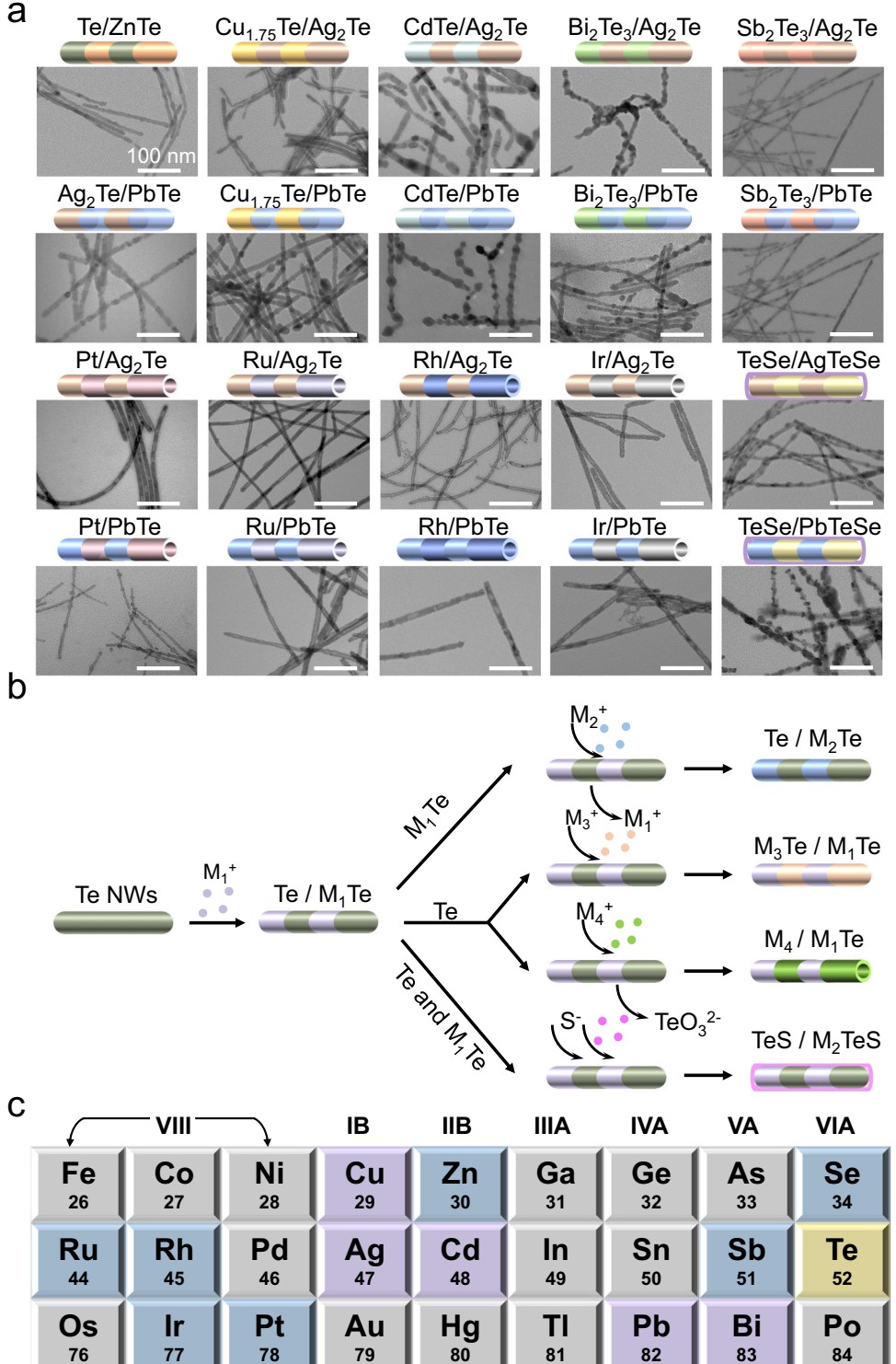

**Fig. 4 | The subsequent chemical transformations and the obtained 1D SHs.**
**a** Typical TEM images of as-transformed SHs with the corresponding diagram on the top. The scale of each image is identical. **b** Schematic illustration of the chemical conversion from Te/M$_1$Te template, showing controllable reactions with the residual Te, the generated M$_1$Te, and both. **c** Overview of the resulting 2$^{nd}$ (highlighted in purple) and 3$^{rd}$ generations (highlighted in indigo) from the initial 1$^{st}$ Te template (highlighted in yellow).

In addition, NW-NT SHs are created through the nanoscale Kirkendall effect. Te/Ag$_2$Te evolved into M$_4$Te/Ag$_2$Te with the moderate precious metal precursor M$_4$ (M$_4$=Pt, Ru, Rh, Ir)[43], in which alternate distribution of Ag$_2$Te NW and M$_4$Te NT is realized. PtTe/Ag$_2$Te, RuTe/Ag$_2$Te, RhTe/Ag$_2$Te and IrTe/Ag$_2$Te NW-NT SHs are shown in Fig. 4a. Suppl. Figs. 35–38, demonstrate the coexistence of solid and hollow segments in such a single 1D nanostructure. PtTe/PbTe, RuTe/PbTe, RhTe/PbTe and IrTe/PbTe NW-NT SHs are also obtained by employing Te/PbTe as templates. The detailed characterizations are provided in the Suppl. Figs. 39–42. It is worth pointing out that the NW-NT SHs are prepared by a simple solution-phase synthesis. Besides, selenylation reaction can also be evoked with G.2 templates (Suppl. Fig. 43). The

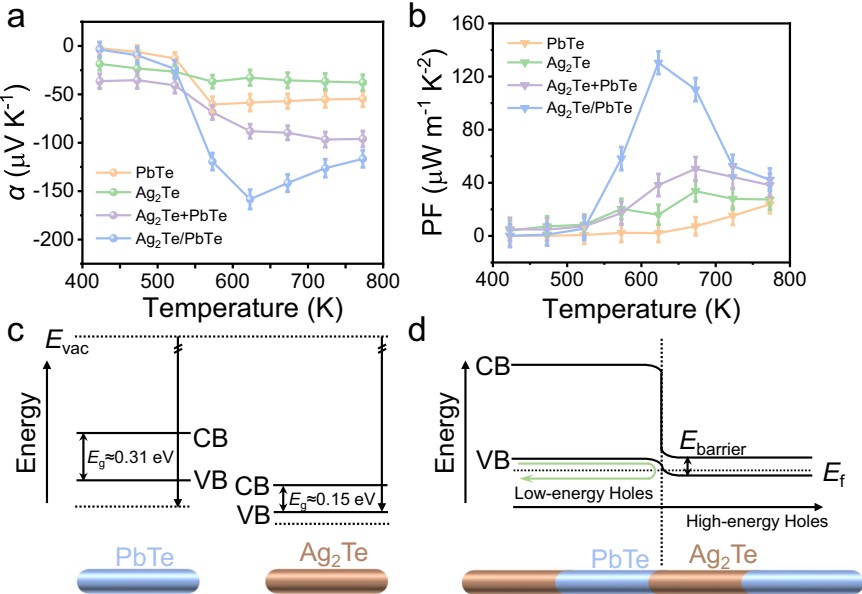

**Fig. 5 | The thermoelectric performance and band structure of Ag$_2$Te/PbTe 1D SHs. a** Seebeck coefficient. **b** Power factor of Ag$_2$Te/PbTe 1D SHs compared with PbTe NWs, Ag$_2$Te NWs, and Ag$_2$Te + PbTe NW mixture between 425 and 775 K. For calculation, the Seebeck coefficient and power factor were measured three times.

**c** Band structure in PbTe NWs and Ag$_2$Te NWs before contact. **d** Equilibrium band alignment in Ag$_2$Te/PbTe 1D SHs. The black horizontal arrow indicates the direction of carrier transport in the Ag$_2$Te/PbTe heterogeneous structure. Source data are provided as a Source Data file.

overall diameter of the resulting TeSe/AgTeSe SHs increases resulting from Se insertion and coating. The selenylation can also be observed when Te/PbTe SHs serves as templates (Suppl. Fig. 44).

The chemical transformation from primary Te NW templates to segmented nanostructures is summarized in Fig. 4b. Considering the under-stoichiometric reaction and reduction potential sequences, Te/M$_1$Te SHs can be obtained with metal precursor added. The subsequent conversion can be performed selectively, *e.g.*, cation exchange with the generated M$_1$Te, combination, and substitution with the residual Te or reaction with both. Thus 4 SHs can be prepared, *i.e.*, Te/M$_2$Te NW, M$_3$Te/M$_1$Te NW, M$_4$/M$_1$Te NW-NT and TeS/M$_2$TeS NW. Figure 4c shows a schematic of the periodic table, highlighting the chemical post-transformation library of all 25 G.2 (purple) and G.3 (yellow) SHs with 13 elements involved.

## Thermoelectric performances of Ag$_2$Te/PbTe SHs

As we known, elemental semiconducting tellurium and telluride family show high thermoelectric figure of merit from 300 to 700 K[45,46]. Ag$_2$Te/PbTe SHs were selected as a model to conceptually demonstrate the properties of axial SHs. The as-synthesized SHs were treated with hydrazine/ethanol mixture to remove surfactants and then hot pressed into millimeter-thick plates. Compared to the low/medium temperature window of PbTe-based thermoelectric nanomaterial, the Ag$_2$Te/PbTe SHs show a seebeck coefficient of 159 μV K$^{-1}$ at 625 K (Fig. 5a). Moreover, thanks to the high conductivity of Ag$_2$Te segment (Suppl. Fig. 45a), the power factor of Ag$_2$Te/PbTe SHs reaches 130 μW m$^{-1}$ K$^2$ (Fig. 5b). It is worth mentioning that Ag$_2$Te/PbTe SHs exhibit higher power factor in comparison with PbTe-Ag$_2$Te NW mixture due to the heterogeneous interfaces. As shown in Suppl. Fig. 45b, functioning up to 700 K and the easy large-scale preparation lay the foundation for their practical application in thermoelectricity. The carrier traveling axially in SHs, especially the hopping from the Ag$_2$Te into PbTe, will encounter a mismatch in energetic states at the heterogeneous interface. Based on the valence band maxima and band gaps (Suppl. Fig. 46), the electronic band structure of Ag$_2$Te/PbTe SHs vs normal hydrogen electrode (NHE) was determined in Fig. 5c, d. The valence band bending at the interface introduces an energetic barrier

that preferentially scatters low energy carriers. Heterogeneous interfaces in SHs enable the rational engineering of carrier filtering via modulating the carrier dynamics at interfaces.

## Discussion

By incorporating the general under-stoichiometric reaction in wet-synthesis and subsequent chemical transformations, we realize a library of SHs, including 25 NW-NW and NW-NT nanostructures. Mechanical simulations suggest the three-stage evolution process, *i.e.*, island generation, stripe penetration and segment ordering, ruling out the possibility of alloy and core-shell formation. Mechanical calculations also demonstrate the stress-induced ordering mechanism determined by elastic energy minimization, rendering the interface movement and segment ordering. This provides a broad tool set for the synthesis of well-controlled axially segmented heterostructures, expanding the palette of material selection with obvious implications for phonon transport and thermoelectric applications. It deserves to be mentioned that this ordering investigation also contributes to understanding the formation of periodic nanostructures.

## Methods

### Synthesis of Te/PbTe and Te/CdTe SHs

Freshly prepared Te NWs were added with moderate Pb(NO$_3$)$_2$ (99%) or CdCl$_2$ · 2.5H$_2$O (99%) and stirred vigorously at room temperature for 4 h[47]. Then the mixture solution was sealed and maintained at 100 °C and 140 °C for 12 h, respectively. The products were precipitated and washed with ethanol (99.7%) for future characterizations and transformations.

### Synthesis of Te/Bi$_2$Te$_3$ SHs

0.40 mmol clean Te NWs were dispersed in 30.0 mL polyethylene glycol (TEG, 99.0%), into which 0.60 g poly(vinylpyrrolidone) (PVP, K-30, 99.8%), moderate Bi(NO$_3$)$_3$ · 5H$_2$O (99%), 1.0 mL N$_2$H$_4$ · H$_2$O (85 wt% water solution) and 20.0 mg NaOH (96%) were added in sequence. Subsequently, the mixture was transferred into a 50.0 mL three-neck flask equipped with a programmed heater. The solution was heated from room temperature to 200 °C for 20 min and kept at 200 °C for

another 20 min under magnetic stirring and $N_2$ protection[48] (Suppl. Data 1). The products were precipitated and washed with ethanol for future characterizations.

### Synthesis of Te/Cu₁.₇₅Te SHs

0.10 mmol clean Te NWs were dispersed in 30.0 mL ethylene glycol (EG, 99%) and stirred vigorously at room temperature for complete redispersion. Moderate $Cu(NO_3)_2 \cdot 3H_2O$ (99%) and 1.0 mL (1.89 M) ascorbic acid ($V_c$, 99.7%) were added in order and keep stirring for another 2 h[49]. The products were precipitated and washed with ethanol for future characterizations.

### Synthesis of Te/ZnTe SHs

10.0 mL Te/Ag₂Te segmented NWs was washed with ethanol and redispersed in 30.0 mL methanol ($CH_3OH$, 99.7%). Moderate $Zn(NO_3)_2$ (99%) in $CH_3OH$ was added into the suspension, which was then exposed in a water bath at 50 °C under vigorous stirring. 10 min later, 0.20 mL of tributyl phosphate (TBP, 99%) was injected quickly[47]. The cation-exchange reaction was so fast that the color of the solution changed from dark brown to light red in a few minutes. The products were precipitated and washed with ethanol for future characterizations.

### Synthesis of Cu₁.₇₅Te/Ag₂Te, Bi₂Te₃/Ag₂Te, CdTe/Ag₂Te and Sb₂Te₃/Ag₂Te SHs

The preparation of Cu₁.₇₅Te/Ag₂Te, Bi₂Te₃/Ag₂Te, CdTe/Ag₂Te and Sb₂Te₃/Ag₂Te SHs were carried out based on the synthesis of Te/Cu₁.₇₅Te, Te/Bi₂Te₃, Te/CdTe and Te/Sb₂Te₃ SHs with Te/Ag₂Te SHs as template.

### Synthesis of TeSe/AgTeSe SHs

0.10 mmol clean Te/Ag₂Te segmented NWs was redispersed in 30.0 mL $H_2O$. Selenium precursor was prepared by dissolving 31.6 mg selenium powder (Se, 99.9%) in 4.0 mL $N_2H_4 \cdot H_2O$ (85 wt% water solution). After slowly adding moderate Se precursor to the Te/Ag₂Te solution, the mixture was aged at 80 °C for 12 h[34].

### Synthesis of Pt/Ag₂Te, Ru/Ag₂Te, Rh/Ag₂Te and Ir/Ag₂Te SHs

2.0 mL Te/Ag₂Te segmented NWs was washed with ethanol and redispersed in 15.0 mL EG. After dispersed homogeneously under vigorous stirring, moderate $H_2PtCl_6$ (99%), $RuCl_3 \cdot H_2O$ (38-40% Ru), $RhCl_3$ (38-40% Rh), and $IrCl_3$ (99.8%) in EG were added and kept stirring at room temperature for another 20 min. Then, the mixture was sealed and maintained at 160 °C for 6 h[50]. The products were precipitated and washed with ethanol for future characterizations.

### Synthesis of Ag₂Te/PbTe, Cu₁.₇₅Te/PbTe, Bi₂Te₃/PbTe, CdTe/PbTe and Sb₂Te₃/PbTe SHs

Ag₂Te/PbTe, Cu₁.₇₅Te/PbTe, Bi₂Te₃/PbTe, CdTe/PbTe, and Sb₂Te₃/PbTe SHs were prepared by synthetic methods of Te/Ag₂Te, Te/Cu₁.₇₅Te, Te/Bi₂Te₃, Te/CdTe and Te/Sb₂Te₃ SHs with small modifications of Te/PbTe SHs as template.

### Synthesis of TeSe/PbTeSe SHs

The synthesis of TeSe/PbTeSe SHs was performed in accordance with that of TeSe/AgTeSe SHs. And the Te/PbTe SHs were chosen as the template.

### Synthesis of Pt/PbTe, Ru/PbTe, Rh/PbTe and Ir /PbTe SHs

These SHs were synthesized according to the procedures of Pt/Ag₂Te, Ru/Ag₂Te, Rh/Ag₂Te and Ir/Ag₂Te SHs by replacing the Te/Ag₂Te segmented NWs template with Te/PbTe segmented NWs.

### Sample characterizations

Transmission electron microscopy (TEM) analysis was conducted with a Hitachi H7650 TEM operating at 100 kV. High-resolution transmission electron microscopy (HRTEM) measurements were carried out with JEOL-2010F at an acceleration voltage of 200 kV. Energy-dispersive X-ray spectrometer (EDX) and element mapping data were collected using OXFORD INCA x-sight 7421 attached to the JEOL-2010F TEM. XRD patterns were obtained with a Philips X'Pert Pro Super X-ray diffractometer equipped with graphite monochromatized Cu-K$\alpha$ radiation ($\lambda$=1.54178 Å). X-ray photoelectron spectroscopy (XPS) data were gathered using an ESCALAB-MKII X-ray photoelectron spectrometer with Mg K$\alpha$ radiation as an exciting source (Mg K$\alpha$ h$\nu$=1253.6 eV). The UV–vis absorption spectra and UV–vis–NIR diffuse reflectance spectra were obtained by a Shimadzu UV-2600 spectrometer. Raman spectra analysis was performed by using a LABRAM-HR confocal laser micro-Raman spectrometer with a wavelength of 532 nm. ICP analyses were carried out with Optima 7300 DV instrument. Photoluminescence emission spectra were collected with Fluorolog-3-Tou spectrometer (Jobin Yvon Inc.) with nanostructures dispersed in ethanol at room temperature. Synchrotron radiation photoemission spectroscopy (SRPES) was performed at the Catalysis and Surface Science Endstation in National Synchrotron Radiation Laboratory (NSRL), Hefei.

### In-situ TEM characterizations

The in-situ TEM visualization is carried out in a JEOL JEM2100 (JEOL, Tokyo, Japan) operated at 200 kV equipped with a liquid flow TEM holder Poseidon 500 (Protochips, North Carolina, USA). The liquid cell is made up of two silicon nitride chips and a perfluoroelastomer sealing ring around them (550 × 20 μm² for window, 50 nm for gap). Before loading, surface cleaning of chips by acetone and methanol each for 2 min was carried out to sweep away the photoresist. After this, the chips were placed in an oxygen plasma cleaner (Gatan, Model 950) for 5 min to further remove the residual organic contaminants. 1-2 μL of Te NW solution was dropped onto the clean chip carefully, which was covered by another larger chip. Before inserting into the TEM, pre-vacuum-pumping of the cell was completed in a home-made vacuum pump. A syringe pump (Harvard Apparatus, Pump 11 Elite) was used to control the Ag⁺ liquid flow rate at 100 μL h⁻¹. Cantega G2 camera (Olympus, Tokyo, Japan) was used to record the real-time movies.

### Phase field model

A continuum phase field model[51–55] is developed for investigating the formation of Te/Ag₂Te ordered SHs when Te NWs are exposed in a solution with Ag⁺. We construct Te NWs with the surrounding Ag⁺ solution as a computational system, with two order parameters $c$ and $\varphi$ introduced. $c$ represents the concentration of Ag⁺. The high-concentration region represents Ag₂Te phase while the low concentration region represents Te phase. $\varphi = 1$ and $\varphi = 0$ represent the solid phase and the ambient solution, respectively. The interface between the solid and the solution has a $\varphi$ between 0 and 1. It is worth mentioning that stresses only exist in the solid phase. The three stages of Te/Ag₂Te ordered SHs are defined as follows. Firstly, Ag⁺ ions gather at the defective positions randomly distributed on the surface of Te NWs and diffuse into Te NWs to form Ag₂Te islands. Secondly, Ag₂Te islands grow faster in the radial direction and span the entire diameter, forming poorly ordered structure. Finally, the nearby Ag₂Te segments attract each other and merge into one segment, while the distant Ag₂Te segments repel each other, forming well-ordered superlattice.

### Calculation of Ag concentration

In the first stage, we assume that Ag⁺ tend to gather into the defects of Te NWs and impose uneven boundary condition with smooth boundary method. The unit normal vector of the interface, pointing to the solid, is given by $\nu = \nabla\varphi/|\nabla\varphi|$, through which the constant ion current can be embedded into the Cahn-Hilliard equation[53,54]. We

finally get the evolution equation as follows

$$\rho\frac{\partial c}{\partial t} = \frac{1}{\varphi}\left[\nabla\left(M\nabla\left(\frac{\partial f_c}{\partial c} - \kappa\nabla^2 c + \frac{\partial f_e}{\partial c}\right)\right)\right] + \frac{1}{\varphi}|\nabla\varphi|I \quad (1)$$

where $\rho$ is the number of $Ag^+$ per unit volume and $M$ is the mobility. $f_c$ is the chemical free energy density, $\kappa$ is the gradient coefficient, for simplicity, it is assumed to be a constant here. $f_e = \sigma_{ij}\varepsilon_{ij}^e/2$ is the elastic strain energy density, which is caused by lattice parameter change due to the intercalation of $Ag^+$. $\sigma_{ij}$ and $\varepsilon_{ij}^e$ are stress and elastic strain, respectively, which will be detailed below. $I$ represents the ion flux intercalated into nanowire from solution. The expression of chemical-free energy density $f_c$ is

$$f_c = \varphi f_s + (1 - \varphi)f_l \quad (2)$$

where $f_s(c) = \rho\omega c^2(1-c)^2$ represents the chemical free energy density of solid phase. $f_l(c) = \rho\omega(0.4 - c)^2$ is the chemical free energy density of the solution, which are given linear weighting by the phase fraction field. Substituting Eq. (2) into Eq. (1), the evolution of the concentration field is governed by the following Cahn-Hilliard equation.

$$\rho\frac{\partial c}{\partial t} = \frac{1}{\varphi}\nabla\left\{M\varphi\nabla\left(\varphi\frac{\partial f_s}{\partial c} + (1-\varphi)\frac{\partial f_l}{\partial c} - \kappa\nabla^2 c + \frac{\partial f_e}{\partial c}\right)\right\} + \frac{1}{\varphi}|\nabla\varphi|I \quad (3)$$

Although the interfacial energy is assumed to be isotropic in the original governing equation, it can be extended to anisotropic interfacial energy by the following equation.

$$\rho\frac{\partial c}{\partial t} = \frac{1}{\varphi}\left[\nabla\left(M\varphi\cdot\nabla\left(\frac{\partial f_c}{\partial c} - \nabla(\vec{\kappa}\nabla c) + \frac{\partial f_e}{\partial c}\right)\right)\right] + \frac{1}{\varphi}|\nabla\varphi|I \quad (4)$$

where $\vec{\kappa} = [\kappa_x, 0, 0, \kappa_y]$, the ratio $\kappa_x/\kappa_y$ can be used to characterize the degree of anisotropy of the interfacial energy.

Further, the phase field model can be modified to investigate the effects of interfacial diffusion and surface diffusion on the inclusion growth kinetics in the nanowire. It is conventionally assumed that interfacial/surface diffusion occurs more rapidly than bulk diffusion. The bulk diffusion coefficient and surface diffusion coefficient of the nanowire are denoted as $D_b$ and $D_s$, respectively. For simplification, we employed concentration-dependent bulk diffusion coefficient $D_b c(1-c)$ to elucidate the behavior of interfacial diffusion. The governing equation when considering the interfacial diffusion can be written as

$$\rho\frac{\partial c}{\partial t} = \frac{1}{\varphi}\left\{\nabla\left[\frac{D_b c(1-c)}{k_B T}\varphi\cdot\nabla\left(\frac{\partial f_c}{\partial c} - \kappa\nabla^2 c + \frac{\partial f_e}{\partial c}\right)\right]\right\} + \frac{1}{\varphi}|\nabla\varphi|I \quad (5)$$

Based on the surface diffusion model[51], the governing equation considering the surface diffusion can be written as

$$\rho\frac{\partial c}{\partial t} = \frac{D_b}{\varphi k_B T}\nabla\cdot\left[\varphi\nabla\left(\frac{\partial f_c}{\partial c} - \kappa\nabla^2 c + \frac{\partial f_e}{\partial c}\right)\right] + \frac{|\nabla\varphi|}{\varphi}\left[\nabla_s\cdot\frac{\lambda D_s}{k_B T}\nabla_s\left(\frac{\partial f_c}{\partial c} - \kappa\nabla^2 c + \frac{\partial f_e}{\partial c}\right) + I - \rho\lambda\frac{\partial c}{\partial t}\right] \quad (6)$$

where $\lambda$ is the characteristic thickness of the surface zone, $\nabla_s$ is the surface gradient operator.

## Calculation of elastic energy
It should be noted that the elastic modulus is quite different between the solid phase and solution, and the modulus of the NW is dependent on the Ag concentration. In addition, the elastic strain induced by the

lattice mismatch of Ag in Te NWs is inhomogeneous. Thereby, the computational system composed by the NW and the surrounding solution is an elastically and structurally inhomogeneous system, which is clearly defined in the phase field microelasticity theory[52]. The main formula for the calculation of the stresses and strains is listed below:

In this theory, the elastic strain $\varepsilon_{ij}^e(\mathbf{r})$ can be expressed by the total strain $\varepsilon_{ij}(\mathbf{r})$ and the chemical eigenstrain $\varepsilon_{ij}^c(\mathbf{r})$

$$\varepsilon_{ij}^e(\mathbf{r}) = \varepsilon_{ij}(\mathbf{r}) - \varepsilon_{ij}^c(\mathbf{r}) \quad (7)$$

and the chemical eigenstrain is described as

$$\varepsilon_{ij}^c(\mathbf{r}) = \beta(c(\mathbf{r}) - c_0)\delta_{ij} \quad (8)$$

where $\beta$ is the expansion coefficient, $c_0$ is the reference concentration and $\delta_{ij}$ is Kronecker delta function. According to Hooke's law, the elastic stress is expressed as

$$\sigma_{ij}(\mathbf{r}) = C_{ijkl}(\mathbf{r})\left[\varepsilon_{kl}(\mathbf{r}) - \varepsilon_{kl}^c(\mathbf{r})\right] \quad (9)$$

Here $C_{ijkl}(\mathbf{r})$ is the position-dependent modulus, which can be expressed as $C_{ijkl}(\mathbf{r}) = \varphi[0.593c(\mathbf{r}) + 0.407] \times C_{ijkl}^0$. When the domain parameter equals 0 ($\varphi = 0$), it indicates that the modulus of the solution is zero. When the domain parameter equals 1 ($\varphi = 1$) and the concentration equals 1 ($c = 1$), the modulus of $Ag_2Te$ is $C_{ijkl}^0$. When the domain parameter equals 1 ($\varphi = 1$) and the concentration equals 0 ($c = 0$), the modulus equals Te NW modulus ($0.407 C_{ijkl}^0$). We can transform the inhomogeneous system into an equivalent elastically homogeneous system by introducing the virtual eigenstrain $\varepsilon_{ij}^0(\mathbf{r})$

$$C_{ijkl}^0\left[\varepsilon_{kl}(\mathbf{r}) - \varepsilon_{kl}^0(\mathbf{r})\right] = \left[C_{ijkl}^0 - \Delta C_{ijkl}(\mathbf{r})\right]\left[\varepsilon_{kl}(\mathbf{r}) - \varepsilon_{kl}^c(\mathbf{r})\right] \quad (10)$$

where $C_{ijkl}^0$ is the reference moduli and $\Delta C_{ijkl}(\mathbf{r})$ is the moduli difference with the reference modulus of $Ag_2Te$. Equation (10) establishes an equivalent relationship between the original elastically inhomogeneous system and the equivalent elastically homogeneous system. The virtual eigenstrain $\varepsilon_{ij}^0(\mathbf{r})$ is governed by a time-dependent Ginzburg-Landau-type equation[54]

$$\frac{\partial\varepsilon_{ij}^0(\mathbf{r},t)}{\partial t} = -L_{ijkl}\frac{\delta E^{\text{inhom}}}{\delta\varepsilon_{kl}^0(\mathbf{r},t)} \quad (11)$$

where $t$ is virtual time, $L_{ijkl}$ is the kinetic coefficient ($L_{ijkl} = L\delta_{ik}\delta_{jl}$) and $E^{\text{inhom}}$ is the elastic energy of the elastically and structurally inhomogeneous system[55]

$$E^{\text{inhom}} = \frac{1}{2}\int_V\left[C_{ijmn}^0\Delta C_{mnpq}^{-1}(\mathbf{r})C_{pqkl}^0 - C_{ijkl}^0\right]\left[\varepsilon_{ij}^0(\mathbf{r}) - \varepsilon_{ij}^c(\mathbf{r})\right]\left[\varepsilon_{kl}^0(\mathbf{r}) - \varepsilon_{kl}^c(\mathbf{r})\right]dV$$
$$+ \frac{1}{2}\int_V C_{ijkl}^0\varepsilon_{ij}^0(\mathbf{r})\varepsilon_{kl}^0(\mathbf{r})dV + \frac{V}{2}C_{ijkl}^0\bar{\varepsilon}_{ij}\bar{\varepsilon}_{kl} - \bar{\varepsilon}_{ij}\int_V C_{ijkl}^0\varepsilon_{kl}^0(\mathbf{r})dV$$
$$- \frac{1}{2}\int_{|k|\neq 0}n_i\sigma_{ij}^0(\mathbf{k})\Omega_{jk}(\mathbf{n})\sigma_{kl}^{0\prime}(\mathbf{k})n_l e^{i\mathbf{kr}}\frac{d^3k}{(2\pi)^3} \quad (12)$$

where $\bar{\varepsilon}_{ij}$ is average strain, $\widetilde{\sigma}_{ij}^0(\mathbf{k})$ is the Fourier transform of $\sigma_{ij}^0(\mathbf{r})$ and $\sigma_{ij}^0(\mathbf{r})$ is the eigenstress that is expressed as $\sigma_{ij}^0(\mathbf{r}) = C_{ijkl}^0\varepsilon_{ij}^0(\mathbf{r})$. $\widetilde{\sigma}_{kl}^{0\prime}(\mathbf{k})$ is the complex conjugate of $\widetilde{\sigma}_{kl}^0(\mathbf{k})$, $\Omega_{jk}(\mathbf{n})$ is the Green function tensor and $\Delta C_{mnpq}^{-1}(\mathbf{r})$ is the inverse of $\Delta C_{mnpq}(\mathbf{r})$. Once $\varepsilon_{ij}^0(\mathbf{r})$ is obtained from Eq. (11) and Eq. (12), the stresses $\sigma_{ij}(\mathbf{r})$ is can be obtained by

$$\sigma_{ij}(\mathbf{r}) = \frac{1}{2}C_{ijkl}(\mathbf{r})\left\{\int_{|k|\neq 0}\left[n_i\Omega_{jk}(\mathbf{n}) + n_j\Omega_{ik}(\mathbf{n})\right]C_{klmn}^0\varepsilon_{ij}^0(\mathbf{k})n_l e^{i\mathbf{kr}}\frac{d^3k}{(2\pi)^3} + \bar{\varepsilon}_{ij} - \varepsilon_{ij}^c(\mathbf{r})\right\} \quad (13)$$

and the elastic energy density $f_e$ is expressed as

$$f_e = \frac{1}{2}\sigma_{ij}(\mathbf{r})\varepsilon_{ij}^e(\mathbf{r}) \tag{14}$$

## Numerical simulation

To perform the numerical simulations in the method part, we introduce the following dimensionless parameters

$$f_c^* = \frac{f_c}{k_B T\rho}, \qquad \omega^* = \frac{\omega}{k_B T}, \qquad \kappa^* = \frac{\kappa}{k_B T\rho l^2}, \qquad f_e^* = \frac{f_e}{k_B T\rho},$$

$$W^* = \frac{\int f_e dV}{k_B T} \quad C_{ijkl}^0{}^* = \frac{C_{ijkl}^0}{G_{Ag_2Te}}, \qquad \sigma_{ij}^* = \frac{\sigma_{ij}}{G_{Ag_2Te}}, \qquad \Omega_{ij}^* = \Omega_{ij}G_{Ag_2Te}, \tag{15}$$

$$I^* = \frac{t_0}{\rho l}I, \qquad t^* = \frac{t}{t_0}, \qquad L^* = LG_{Ag_2Te}t_0, \qquad M^* = \frac{k_B T t_0}{l^2}M$$

where $k_B$ is Boltzmann constant and $T$ is the absolute temperature. $l$ is the characteristic length of the NW and $t_0 = l^2/D$ is the characteristic time. We can obtain dimensionless evolution equations for concentration and virtual eigenstrain $\varepsilon_{ij}^0(\mathbf{r})$

$$\frac{\partial c}{\partial t^*} = M^*\frac{1}{\varphi}\left[\nabla^*\left(\varphi\nabla^*\left(\frac{\partial f_c^*}{\partial c} - \kappa^*\nabla^{*2}c + \frac{\partial f_e^*}{\partial c}\right)\right)\right] + \frac{1}{\varphi}\left|\nabla^*\varphi\right|I^* \tag{16}$$

$$\frac{\partial \varepsilon_{ij}^0(\mathbf{r}^*,t^*)}{\partial t^*} =$$
$$L^* C_{ijkl}^0{}^* \left\{ \frac{1}{2}\int\limits_{|\mathbf{k}|\neq 0}\left[n_k\Omega_{lm}^*(\mathbf{n}) + n_l\Omega_{km}^*(\mathbf{n})\right]\sigma_{mn}^0{}^*(\mathbf{k})n_n e^{i\mathbf{kr}^*}\frac{d^3k}{(2\pi)^3} - \varepsilon_{kl}^0(\mathbf{r}^*) + \bar{\varepsilon}_{kl}^0(\mathbf{r}^*)\right\} \tag{17}$$

Numerical simulations are performed using software MATLAB. Fast Fourier transform algorithm is employed to solve the governing equations. The physical parameters selected from previously published works are listed in Suppl. Table 2. We set the same dimensionless time step $\Delta t^* = 0.005$ for the evolution of concentration and virtual eigenstrain. Based on the Einstein relation $M = D/(k_B T)$, the value for the mobility $M$ in Te/Ag$_2$Te system is calculated as $M = 1.2\times 10^7\,m^2j^{-1}\,s^{-1}$. The kinetic coefficient $L$ governs the artificial evolution rate of virtual eigenstrain, which is chosen as $L = 1.85\times 10^{-10}\,m^3 j^{-1}s^{-1}$. With the normalized parameters in Eq. (15), the dimensionless kinetic coefficients are calculated as $L^* = 2$ and $M^* = 1$.

## Evolution of elastic energy with segment spacing

In the third stage, we focus on the energy change with the segment spacing, which is the driving force for the close-attract far-repel of the inclusions. First, we construct two Ag$_2$Te segments with a constant spacing $H$ and keep the total length of the two Ag$_2$Te segments constant. When we change the difference between the lengths of the two Ag$_2$Te segments $(h_1 - h_2)$, we can obtain the variation of the elastic energy as a function of $h_1 - h_2$. Second, we construct one Ag$_2$Te segment and two Te segments. Similarly, we set the length Ag$_2$Te segment $(h)$ to be constant and keep the sum of the two spacings of Te segments $H_1$ and $H_2$ unchanged. Then, we alter the length difference of two Te segments $H_1 - H_2$, and calculate the elastic energy under different $H_1 - H_2$.

## Simulations of Te/Cu$_{1.75}$Te and Te/PbTe SHs

Our phase field model can be applied to simulate the formation of Te/Cu$_{1.75}$Te and Te/PbTe SHs. The evolution equations and numerical simulation procedures are the same as those of Te/Ag$_2$Te system, only Young's modulus, Poisson's ratio, particle number per unit volume, expansion coefficient and diffusion coefficient need to be replaced by

the corresponding values of each system. Then, we can get the results of dimensionless elastic energy varying with dimensionless segment length difference of Te/PbTe and Te/Cu$_{1.75}$Te systems, respectively.

## Reporting summary

Further information on research design is available in the Nature Portfolio Reporting Summary linked to this article.

## Data availability

The data that support the findings of this study are available from the corresponding authors upon request. Source data are provided with this paper.

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

## Acknowledgements

This work was supported by the National Natural Science Foundation of China (Grants 22293044, S.-H.Y., 22175164, J.-W.L., 12025206, Y.N., 22005285, Q.-X.C.), the National Key Research and Development Program of China (2021YFA0715700), Strategic Priority Research Program of the Chinese Academy of Sciences (XDB0450402, S.-H.Y., XDB0620101, Y.N.), the Major Basic Research Project of Anhui Province (2023z04020009), the Fundamental Research Funds for the Central Universities (WK2100000005, WK2090050046, Y.-Y.L.), the National Postdoctoral Program for Innovative Talents (Grant BX20200316, Q.-X.C.), and the China Postdoctoral Science Foundation (2020M671869, Q.-X.C.). This work was partially carried out at the USTC Center for Micro and Nanoscale Research and Fabrication, also at the Instruments Center for Physical Science, University of Science and Technology of China.

## Author contributions

S.-H.Y., and J.-W.L. conceived the idea and designed the experiments, and supervised the research. S.-H.Y., J.-W.L., and Y.N. supervised the research. Q.-X.C. carried out the experiments and characterizations. Y.-Y.L., Yang Y.,

and L.-G.C carried out the phase field simulations. Y.-Y.L., Y.Y., L.-G.C, and Y. N. performed the theoretical analyses. Y.L., Yuan Y, and Z.H. participated in data analysis. Q.-X.C., Y.-Y.L., Y. N., J.-W.L., and S.-H.Y. analyzed the data and co-wrote the manuscript. All authors analyzed and discussed the results.

## Competing interests

The authors declare no competing interests.
