## [Peer Review File · Nature Communications]

Stress-induced ordering evolution of 1D segmented heteronanostructures and their chemical post-transformationsReviewers' Comments:

Reviewer #1:

Remarks to the Author:

This is a very nice article that describes multiphase evolution in 1D systems driven by compositional gradients. The experimental realization of these structures spans multiple systems, and the authors have used continuum phase-field models that combine the thermodynamics, kinetics and mechanics that drives the evolution.

While the article advances the field of compositional modulated structures and is certainly innovative in the systems that have been realized, there are several issues that I would like to see addressed before it becomes suitable for publication in Nature Communications:

Major comments:

- 1) The authors describe the anisotropic (radial/axial) growth in terms of anisotropy in stress development within the island as it grows into the 1D wire. The near-equilibrium configuration of island is determined in part by the balance of the interfaces that bound it. What is effect of the interfacial energy balances and their anisotropy?
- 2) The authors have not addressed the issue of interfacial and surface diffusion and interfacial kinetics in the presence of deposition flux, and its coupling with the stress (diffusion potential, if you will) on the evolution. With their nice phase field model, they should be able to address these issues with rough parametric exploration of the diffusivities.
- 3) The near-attracts far-repels paradigm may not be universal. The authors need to do a parametric study with varying mismatch strains to show that this is not system specific, rather it extends to all systems. The analysis for AgTe system seems robust, but I am not convinced that this is a universal result and there may be other kinetic factors in play in different systems. For example, the larger lattice mismatch in Pb-Te system needs to be re-analyzed in terms of additional changes in the kinetic and thermodynamic parameters in that system.

Additionally, is it possible for the authors to show a scaling analysis which captures the length-scale dependence of the inter-segment interaction, the parameters it depends on etc., thereby showing clearly how this interaction changes with spacing? This analysis should make it to the main text (not supplemental documents).

- 4) It is very nice to see the evolution of the segmentation in the NW in the experiments (Fig. 1g). Is it possible for the authors to come up with an expression for the time dependence of the spacing between the segments, guided by the model and scaling based theoretical analysis. The time exponent of the size and spacing between the phases would be a very nice result, analogous to Ostwald ripening in the presence of an external flux (although I admit that the analogy is far from perfect).
- 5) Under what parameters do the authors see a transition to the Kirkendall effect? Can they see this transition in the phase-field model? If so, this would be a useful guide for the experiments. In general, the flavor of the article appears to be explaining the experimental observations. I would like the authors to show that the model is predictive, that is can it predict the segmentation in a new system which is then validated by the experiments? I realize that the authors have explored many systems, and perhaps this is simply a matter of reorganizing the presentation of the results.

Minor comments:

- 1) I think the title does not do justice to the article, as it is a bit vague and does not communicate the

focus of the article.

2) Perhaps a better phrase for near-attracts, far-repels? If it is used heavily in prior literature, then its okay.

3) Some words do not fit in well with the sentences (e.g. "vindicated"). I would advice the authors to simplify the language to so that the intent is conveyed as they intended.

Reviewer #2:

Remarks to the Author:

This work highlights an interesting approach to synthesize segmented nanowire heterostructures starting from Te nanowires. The formation of the Te/Ag₂Te segmented heterostructures is discussed in great detail, also with in situ microscopy techniques and is additionally modeled. From these heterostructures, other heterostructures can be generated in a two-step approach. Various groups have previously covered the formation of heterostructures but the mechanism reported in this work appears new. I have the following remarks:

-The whole story hinges on Te nanowires and is therefore doubtful that the findings of the work can be easily extended to other materials.

-The mechanism proposed here should have a strong dependence on the diameter of the wires, but this is a critical parameter that the authors are not exploring. To really prove the mechanism, which should most likely predict different sizes of the segments based on strain balance, depending on wire diameter, it is advisable indeed to experimentally probe wires with different diameters. If this is not done, in my view the overall explanation and model are somewhat speculative.

-The second part of the work is considerably less interesting, as the heterostructures are then transformed into other heterostructures with different compositions following well-known procedures (such as cation exchange).

-In this work the TEM images cover limited areas of each sample. Being TEM a very local technique, one can always scan the grid in a region showing samples that appear to confirm one's hypothesis. Hence for each of the transformations reported here the authors should present a larger set of TEM images, including wide fields of view, supporting their claims.

Reviewer #3:

Remarks to the Author:

Report Attached. My name can be shared with the report

Review of NCOMMS-23-25379: “Ordering evolution enabled a library of one-dimensional segmented heteronanostructures”.

Overall, this manuscript describes interesting synthesis and characterization studies of one-dimensional (1D) heteronanostructures, including nanowire-nanowire (NW-NW) and nanowire-nanotube (NW-NT) materials. Over the past several decades there has been considerable interest in 1D heteronanostructures, often being termed axial or radial superlattices since previous work generally focused on semiconductor-semiconductor and/or semiconductor-metal structures originally studied in the context of planar (2-dimensions, 2D) superlattices. The current focus on a broad range of metal heteronanostructures breaks new ground and can make a unique and important contribution to the literature that this reviewer believes will be interesting to readers of *Nature Comm.* as well as the active field of nanostructure synthesis. Specific strengths and weaknesses of the work are as follows:

1. The general approach described in the manuscript is quite interesting and compliments previous work. For example, there has been considerable effort (related to 2D superlattices) whereby sequential delivery of reactants has been used to prepare a broad spectrum of axial and radial NW superlattices. Also, phase separation strategies trading off thermodynamic and kinetic constraints have also led to heteronanostructures superlattices. The new work might be claimed to be most related to this later area of prior work, but I personally feel, the concept and approach remain quite distinct and original; that is, it should stimulate considerable interest by other researchers to apply this to other materials as well.
2. The TEM data – static images and movies – provide strong evidence supporting the proposed model and successful synthetic realization of the wide-range of heteronanostructures (i.e., Fig. 1, Fig. 4a, supplementary movies). Nevertheless, this data also might be one of the weaker points of the paper, especially within the context of model structures/modeling shown, for example, in Figs. 2 & 3. The hallmark of previous work in the field has been very clear images of the axial and radial superlattice repeats and material boundaries/interfaces often with high-resolution atomic images. While I do not believe the present level of TEM data should disqualify the paper, I would also urge the authors to see whether they can provide sharper data, including movies, that illustrate better 1-2 of heteronanostructures materials.
3. It would be better in this reviewer’s opinion not to somewhat bury the thermoelectric property characterization in Supplementary Figure 44. We agree with the authors that it is important at this stage in the field to demonstrate a potential direction that is enabled by a new nanomaterial, so the effort to characterize thermoelectric properties is appreciated. Nevertheless, I think the paper could be strengthened by addressing this key result (it is in fact a motivation for others why they should be interested in the work if not simply interested in materials synthesis) in considerably more depth. Specific points that the authors should consider might include the following: (a) perhaps it would be more effective to include a clear description of the thermoelectric measurements and results in a figure in the main text (e.g., the last main text figure); (b) it is unclear how the measurements were carried out and this is important to the interpretation (and possible criticism by experts in this active research area). For example, ideally it the most meaningful measurement would be made on *individual* heteronanostructures; yes, they are most challenging but have been achieved by several groups in the field. If the measurements were, however, carried out on ensembles of the heteronanostructures it is unclear whether results reflect the intrinsic properties of the heteronanostructures, the differing nanostructure-to-nanostructure electrical/thermal transport (which would vary with materials due to surface differences) or some combination of the two.

4. Probably least important to this reviewer but a topic that can cause anger is the citations to the literature. I think the current version of the manuscript has a somewhat random choice of citations (this is made difficult by the vastness of the literature), and I would like to respectfully suggest the following to the authors. Cite several of the most early (seminal) NW-NW and NW-NT reports, but also cite reviews from the major groups, including at least one recent review, that have at least partially focused on this subject because these reviews have generally cited comprehensively many more papers than can be cited in an original research publication (this can help to allay disagreement with authors who feel their work should have been cited).

Reviewer: Charles M. Lieber

Reviewer #4:

Remarks to the Author:

The authors synthesize Ag₂Te/Te nanowires with striped periodicity and attribute this to a “near attracts far repels” hypothesis that comes out of phase field modeling. They then apply a range of established chemical transformation reactions to convert this to a library of derivative multi-component nanowires. There are some potentially interesting hypotheses and results here, but I do not recommend the manuscript for publication in Nature Comm. I find inconsistencies between the experimental data and the model and I struggle to link what is claimed as “near attracts far repels” to what is shown experimentally. This was also a very confusing paper to read and I fear that even experts in the field will struggle to comprehend it. I explain more about my rationale below.

Introduction: Some important aspects of the field are highly over-sold and over-hyped. At the same time, other aspects are unrealistically over-simplified. It would be appreciated if the authors could better balance this.

Precedent: Most of what is described for the Ag⁺ reaction with Te nanowires, forming Te/Ag₂Te, seems akin to the initial Science paper on Ag₂Se/CdSe superlattice nanowires by Alivisatos, in terms of the (hypothesized at the time) process and the heterostructured morphology. In the Alivisatos paper, this very similar ordering and bulging was attributed (through computational modeling) to strain. How do the authors reconcile this difference in proposed mechanism?

The model: As I understand it, in this current submission, the authors create a model based on elastic energy, identifying a critical threshold for repulsive vs attractive interactions of the similar segments based on elastic energy (page 8). However, their preceding discussion (page 7) focused on diffusion and migration, yet I do not see (or understand) how diffusion of Ag⁺ through Te (and aspects of structure changes and/or redox chemistry that will be important mechanistically) relates to elastic energy. So unless I am missing something, I fail to see how the experimental data relates to the model that is supposed to connect what is observed to what can be predicted. This is the premise and sales pitch for the paper, and it leaves me confused and feeling like the two are contradictory. Perhaps they are not, but the authors were not able to convince me in the current version of the manuscript.

My proposed alternate explanation: In terms of their “near-attracts far-repels” theory, could that not simply be described, more simply, by minimization of interfacial energies (proxy for strain), which is already well established in the field and that has already been used to predict heterostructuring in metal chalcogenide nanorods? The “near attracts” part means that two interfaces would have to co-exist with a narrow separation, and that would be energetically unfavorable, so they combine. The “far repels” part means that two interfaces are far enough apart to not have to interact, and they remain separated and become located at ideal locations for other reasons (strain minimization?). I am not saying that my proposal is accurate, but rather that it perhaps provides a reasonable alternative to elastic energy that, to my current level of understanding, makes a bit more sense in connecting the mechanistic pieces together. In the end, I am left struggling to figure out how elastic energy is relevant. The authors would have to convince me a bit more to help connect this rationale to the details of the system and reaction.

Chemical transformations: The “chemical transformation” section is already established chemistry that adds materials diversity, but doesn’t relate to their “near attracts far repels” hypothesis. It’s nice and adds interest, but this much larger library of systems doesn’t seem to validate (or refute) their hypothesis as the pathway is different. (The ordering is already there before these transformations are carried out, if I understand correctly.) The actual experimental validation of the main mechanistic point of the paper is therefore quite limited.

Materials characterization: In addition to the concerns mentioned above, I also have significant concerns about materials characterization. For example, in Fig 1, the XRD data does not seem to match what they claim; the authors should take a closer look at XRD interpretation throughout. These

are complicated systems and the XRD data (especially in the SI) seems somewhat poor. Also, the STEM-EDS element map signal in Fig 1 seems quite low overall, especially given the dimensions of the nanowires. And, unless I am missing something, the STEM-EDS data in Fig 1 is all that appears to provide somewhat definitive evidence of possible heterostructuring of the materials claimed; the other images are just TEM/HAADF images, which rely on contrast only and therefore could correspond to other materials (Ag instead of Ag₂Te, etc.). Again, unless I am missing something, the materials characterization data are suggestive, but not fully sufficient, to experimentally validate the heterostructuring and its evolution that ultimately is incorporated into the phase field modeling. The authors should also take a closer look at other aspects of the materials characterization in the paper. For example, in Fig. 4, the data is presented in a way that is too small to be useful; I am not able to see most of the characteristic features that are claimed. Also, the XPS data should be fitted. And in Fig S6, the Ag signal (islands) seems too low to be seen.

Writing: The English/grammar needs to be improved significantly throughout. This is a very complex paper that the authors surely want to be read by researchers across multiple disciplines. Unfortunately, it borders on unreadable at quite a few places. The authors also use incorrect technical terminologies throughout, and should very carefully check this.

The point-to-point answers to the Editor's and the referees' comments

We have carefully considered all valuable criticisms and suggestions from the editor and referees, and have made suitable revision accordingly. For clearness reason, the answers were marked with RED color and started with “**” and the revision parts in the revised manuscript were also marked with RED color.

The point-to-point answers to the referees' questions

We appreciated the critical comments and suggestions from four reviewers. Here, we answered their questions and made suitable revisions as marked by RED color in the following.

Reviewer 1:

This is a very nice article that describes multiphase evolution in 1D systems driven by compositional gradients. The experimental realization of these structures spans multiple systems, and the authors have used continuum phase-field models that combine the thermodynamics, kinetics and mechanics that drives the evolution. While the article advances the field of compositional modulated structures and is certainly innovative in the systems that have been realized, there are several issues that I would like to see addressed before it becomes suitable for publication in Nature Communications:

** Thanks for the professional and nice comments. In this work, we investigated the interesting evolution mechanism of one-dimensional segmented heteronanostructures *via* the continuum phase field models. The phase field method is acknowledged as a powerful computational approach to modeling and predicting the mesoscale morphological and microstructural evolution in materials, especially outstanding in describing the phase transformation coupled with multi-field physics. By using a set of conserved and non-conserved field variables including the thermodynamic, kinetic and mechanic information, the temporal and spatial evolution of a microstructure can be achieved (*Annu. Rev. Mater. Sci.* **2002**, 32, 113–140; *Int. J. Solid Struct.* **2018**, 143,73–83). Based on the phase field model, the formation and ordering mechanism of the segmented heteronanostructures can be interpreted.

Major comments:

1. The authors describe the anisotropic (radial/axial) growth in terms of anisotropy in stress development within the island as it grows into the 1D wire. The near-equilibrium configuration of island is determined in part by the balance of the interfaces that bound it. What is effect of the interfacial energy balances and their anisotropy?

** Thanks for the valuable question. We agree that the near-equilibrium configuration of island is governed by the combination of elastic energy and interfacial energy. The influence of interfacial energy on the inclusion's configuration has been incorporated into our phase field model, which can be seen in the governing equation.

$$\rho \frac{\partial c}{\partial t} = \frac{1}{\varphi} \left[\nabla \cdot \left(M \varphi \cdot \square \left(\frac{\partial f_c}{\partial c} - \kappa \nabla^2 c + \frac{\partial f_e}{\partial c} \right) \right) \right] + \frac{1}{\varphi} |\nabla \varphi| I \quad (1)$$

where ρ is the number of Ag^+ per unit volume and M is its mobility. f_c and f_e are the chemical free energy density and elastic energy density, respectively. κ is the isotropic gradient coefficient. I represents the flux inserted into nanowire. φ is the domain parameter in phase field model.

Although the interfacial energy is assumed to be isotropic in original governing equation, it can be extended to anisotropic interfacial energy by the following equation

$$\rho \frac{\partial c}{\partial t} = \frac{1}{\varphi} \left[\nabla \cdot \left(M \varphi \cdot \square \left(\frac{\partial f_c}{\partial c} - \nabla \cdot (\vec{\kappa} \nabla c) + \frac{\partial f_e}{\partial c} \right) \right) \right] + \frac{1}{\varphi} |\nabla \varphi| I \quad (2)$$

where $\vec{\kappa} = [\kappa_x, 0, 0, \kappa_y]$, the ratio κ_x / κ_y can be used to characterize the degree of anisotropy of interfacial energy.

To clarify the effect of interfacial energy, we first omit the contribution of elastic energy by setting the elastic energy density to be 0. When the ratio κ_x/κ_y is set as 1/4, the simulation results of the growth of Ag_2Te island are shown in New Figure S10a. For simplicity, the simulation results are shown in the middle cross section of nanowire. We can see that the growing Ag_2Te island yields segment in the nanowire, but the island morphology during growth is not consistent with the experiential observation in New Figure S10 in the original manuscript. However, when the ratio is set as $\kappa_x/\kappa_y=4$, the simulation results of the growth of Ag_2Te island are shown in New Figure S10b. In this case, the Ag_2Te island even dose not yield segment in nanowire due to the large ratio of κ_x/κ_y .

New Figure S10. Effect of anisotropic interfacial energy on the growth of Ag_2Te island without considering the stress at different dimensionless times ($t^*=200, 400, 600, 800$). a, The anisotropic interfacial energy parameter $\kappa_x/\kappa_y=1/4$. b, The anisotropic interfacial energy parameter $\kappa_x/\kappa_y=4$.

When the expansion coefficient is set to be 0.02, The Ag_2Te inclusion growth process is simulated for the interfacial energy parameter $\kappa_x/\kappa_y=1/4$ and $\kappa_x/\kappa_y=4$ in Figures R1 and R2, respectively. We can see that the growing Ag_2Te island yields segment when considering the effect of elastic energy, even in the case of the large ratio $\kappa_x/\kappa_y=4$.

Figure R1. The growth of Ag_2Te island when the expansion coefficient is set as 0.02 with the anisotropic interfacial energy parameter $\kappa_x/\kappa_y=1/4$. (a)-(d) The morphology of Ag_2Te island at different dimensionless times ($t^*=100, 400, 700, 1200$). (e)-(h) The distribution of dimensionless hydrostatic stress corresponding to (a)-(d).

Figure R2. The growth of Ag_2Te island when the expansion coefficient is set as 0.02 with the anisotropic interfacial energy parameter $\kappa_x/\kappa_y=4$. (a)-(d) The morphology of Ag_2Te island at

different dimensionless times ($t^*=300, 600, 900, 1500$). (e)-(h) The distribution of dimensionless hydrostatic stress corresponding to (a)-(d).

When the expansion coefficient is set as 0.04, the Ag_2Te inclusion growth process is also simulated as the interfacial energy parameter $\kappa_x/\kappa_y=1/4$ and $\kappa_x/\kappa_y=4$ in New Figure S11, respectively. We can see that the morphology of Ag_2Te inclusion is similar to the experimental observation in Fig. 1h. It can be concluded that during the dynamic process of the island inclusion growth into cylindrical configuration, the stress plays a much more important role than interfacial energy, although interfacial energy has an influence on the dynamics. Nevertheless, its effect is limited.

New Figure S11. Effects of anisotropic interfacial energy and mismatch strain on the growth of Ag_2Te island. a, b, The morphology of Ag_2Te island at different dimensionless times ($t^*=200, 400, 600, 1000$) and the corresponding distribution of dimensionless hydrostatic stress when the anisotropic interfacial energy parameter κ_x/κ_y is set as 1/4, and the expansion coefficient is set as 0.04. c, d, The morphology of Ag_2Te island at different dimensionless times ($t^*=200, 500, 800, 1200$) and the corresponding distribution of dimensionless hydrostatic stress when the anisotropic interfacial energy parameter κ_x/κ_y is set as 4, and the expansion coefficient is set as 0.04.

Further, it is imperative to clarify that the anisotropic stress means that the hydrostatic stress at the front and sides of island is different, specifically tensile stress and compressive stress, respectively. Recalling the stress dependent chemical potential,

$$\mu = \frac{\partial f_c}{\partial c} - \kappa \nabla^2 c - \beta \sigma_h \quad (3)$$

where β is the expansion coefficient and σ_h is hydrostatic stress. It demonstrates that the tensile hydrostatic stress will decrease the chemical potential but compressive hydrostatic stress will increase the chemical potential. Note that the ion favors to migrate into the tensile stress region to decrease the chemical potential. As a result, the tensile hydrostatic stress at the front of island will facilitate the growth of Ag_2Te island into cylinder morphology.

Whether the stress is compressive or tensile at the front and sides of island, it is primarily governed by the expansion coefficient of the inclusion, which is usually independent of the specific

material system. If the expansion coefficient of the island inclusion is positive, the front of island will subject to tensile stress and the sides of island will subject to compressive stress. Note that the presence of cylindrical inclusions is observed in numerous other material systems, which is postulated to be primarily driven by stress-induced radial growth of island.

2. The authors have not addressed the issue of interfacial and surface diffusion and interfacial kinetics in the presence of deposition flux, and its coupling with the stress (diffusion potential, if you will) on the evolution. With their nice phase field model, they should be able to address these issues with rough parametric exploration of the diffusivities.

** Thanks for the helpful suggestion. Based on our prior research (*J. Power Sources* **2021**, 494, 229777), in which the process of interfacial/surface diffusion in a polycrystalline particle is simulated, we can modify the model in this work to investigate the impacts of interfacial and surface diffusion in Te/Ag₂Te nanowire, respectively. It should be pointed out the interfacial diffusion coefficient and surface diffusion coefficient are difficult to be measured and theoretically calculated. Thus, we will conduct rough parametric exploration, as the reviewer suggested.

First, we will elucidate the effect of interfacial diffusion and ignore the effect of surface diffusion. In general, for simplification, it is conventionally assumed that interfacial diffusion occurs more rapidly than bulk diffusion. The diffusion coefficient in the nanowire bulk is denoted as D_b . We employed concentration-dependent bulk diffusion coefficient $D_{bc}(1-c)$ to elucidate the behavior of interfacial diffusion. The governing equation can be written as

$$\rho \frac{\partial c}{\partial t} = \frac{1}{\varphi} \left\{ \nabla \cdot \left[\frac{D_{bc}(1-c)}{k_B T} \varphi \cdot \square \left(\frac{\partial f_c}{\partial c} - \kappa \nabla^2 c + \frac{\partial f_e}{\partial c} \right) \right] \right\} + \frac{1}{\varphi} |\nabla \varphi| I \quad (4)$$

The growth of Ag₂Te island when considering the effect of interfacial diffusion at different dimensionless times is shown in New Figure S12a. We can see that the fast interfacial diffusion process at the interface promotes the growth of islands, thereby accelerating the formation of a cylindrical morphology that traverses the nanowire structure.

New Figure S12. Effects of interfacial diffusion and surface diffusion on the growth of Ag₂Te island. (a) The morphology evolution of Ag₂Te island when considering the effect of interfacial diffusion at different dimensionless times (t* = 200, 400, 800, 1000). (a) The morphology evolution of Ag₂Te island considering fast surface diffusion coefficient ($D_s = 2.5 \times 10^{-9} \text{ cm}^2 \text{ s}^{-1}$) at different dimensionless times (t* = 200, 400, 800, 1000).

Then, to elucidate the effect of surface diffusion, we ignore the effect of interfacial diffusion. We also assumed that the surface diffusion is faster than the bulk diffusion. The governing equation can be written as

$$\rho \frac{\partial c}{\partial t} = \frac{D_b}{\varphi k_B T} \nabla \cdot (\varphi \nabla \mu) + \frac{|\nabla \varphi|}{\varphi} \left(\nabla_s \cdot \frac{\lambda D_s}{k_B T} \nabla_s \mu + I - \rho \lambda \frac{\partial c}{\partial t} \right) \quad (5)$$

where the surface diffusion coefficient is denoted as D_s , λ is the characteristic thickness of the surface zone, ∇_s is the surface gradient operator. μ is the chemical potential, which is given in Eq. (3). The growth of Ag_2Te island when considering the effect of surface diffusion at different dimensionless times is shown in New Figure S12b. It demonstrates that fast surface diffusion may not be conducive to the formation of cylindrical inclusions.

3. The near-attracts far-repels paradigm may not be universal. The authors need to do a parametric study with varying mismatch strains to show that this is not system specific, rather it extends to all systems. The analysis for AgTe system seems robust, but I am not convinced that this is a universal result and there may be other kinetic factors in play in different systems. For example, the larger lattice mismatch in Pb-Te system needs to be re-analyzed in terms of additional changes in the kinetic and thermodynamic parameters in that system.

** Thanks for the professional advice and comments. The elastic energy, which is related to the mismatch strain by,

$$f_e = \frac{1}{2} C_{ijkl}^0 (\varepsilon_{ij}(\mathbf{r}) - \varepsilon_{ij}^*(\mathbf{r})) (\varepsilon_{kl}(\mathbf{r}) - \varepsilon_{kl}^*(\mathbf{r})) \quad (6)$$

The mismatch strain is a dominated factor that determines the ordering morphology. It is a valuable question from the reviewer to discuss about how mismatch strain affects the near-attracts far-repels paradigm. We will answer the question from both thermodynamic and kinetic perspectives.

From the thermodynamic viewpoint, the near-attracts far-repels paradigm of two adjacent cylinder inclusion in a nanowire determined by the elastic interaction is universal. For the given mismatch strain illustrated in New Figure 3e, when the spacing between two inclusions is less than the critical size, decreasing the spacing between them results in a reduction in the system's elastic energy. On the other hand, when the spacing between two inclusions exceeds the critical size, an increase in spacing also leads to a reduction in the system's elastic energy. The elastic energy driven the near-attracts far-repels paradigm is universal. Additionally, in our experimental data, we observed the similar ordering phenomenon in nanowires of other material systems.

New Figure 3e. The calculated dimensionless elastic energy versus ratios of segment separation to the radius (H/a) and segment length to the radius (h/a)

From a dynamic perspective, this ordering process necessitates the migration of guest ion in the host material. Note that the migration of ions requires to overcome energy barriers between two positions. If the mismatch strain is quite small, the driving force will be not enough to drive the spacing change of the inclusion. If the mismatch strain is large enough to overcome the energy barrier of the guest atom migration, the ordering process will occur. The large mismatch strain will facilitate the ordering. As an example, we investigate the interaction between two segments with the initial space $H/a=1$, with the mismatch strain increasing from 0.005 to 0.045, the morphology evolution is shown in New Figure S22. As expected, for the small mismatch strain (0.005), the space

between two segments hardly changes during evolution, because of the small elastic driving force. When the mismatch strain increases, the segments will merge if the initial space is smaller than the critical value. The large mismatch strain facilitates the coalesce of segments due to the large driving force.

New Figure S22. Effect of mismatch strain on the interaction process between two segments at different dimensionless times ($t^*=0, 400, 800, 1200$). a, b, The morphology of Ag_2Te island and the corresponding distribution of dimensionless hydrostatic stress when the mismatch strain is set as 0.005. c, d, The morphology of Ag_2Te island and the corresponding distribution of dimensionless hydrostatic stress when the mismatch strain is set as 0.025. e, f, The morphology of Ag_2Te island and the corresponding distribution of dimensionless hydrostatic stress when the mismatch strain is set as 0.045.

Further, The Pb/Te nanowire with the larger lattice mismatch has been re-analyzed in terms of additional changes in the kinetic and thermodynamic parameters in this system, the relevant parameters are listed in Table S3. The calculated dimensionless elastic energy vs the ratio of segment length to radius (h/a), and the ratio of segment separation to radius (H/a) are shown in New Figure S23a, b. These results are similar to the results in Ag/Te nanowire system.

New Figure S23a, b. Simulated results of Te/PbTe. (a) The calculated dimensionless elastic energy versus the differences in length between adjacent segments. (b) The calculated dimensionless elastic energy versus the ratio of segment length to radius (h/a), and the ratio of segment separation to radius (H/a).

In conclusion, it can be concluded that the near-attracts far-repels paradigm is universal for the nanowire containing cylinder inclusions, under the condition of large mismatch strain.

Additionally, is it possible for the authors to show a scaling analysis which captures the length-scale dependence of the inter-segment interaction, the parameters it depends on etc., thereby showing clearly how this interaction changes with spacing? This analysis should make it to the main text (not supplemental documents).

** Thanks for the comments. The interaction between two neighboring segments is mainly determined by the elastic energy. We have analyzed the relationship between the elastic energy and spacing between the neighboring two segments based on our prior theoretical work (*Appl. Math. Mech.* **2020**, 41, 1461-1478). To ensure the accuracy, the mechanical balance and the boundary conditions should be guaranteed. The relationship between the elastic energy(W) and the spacing(H) is given by,

$$W = \frac{\pi E a^2 \varepsilon^2}{1-\nu} \left\{ \left[P \left(\frac{l_A}{a} \right) + S \left(\frac{l_A}{a}, \frac{l_B}{a}, H \right) \right] l_A + \left[Q \left(\frac{l_B}{a} \right) + T \left(\frac{l_A}{a}, \frac{l_B}{a}, H \right) \right] l_B \right\} \quad (7)$$

where a is the radius of the nanowire. l_A and l_B are the length of the adjacent two inclusions. To derive the analytical solution, the mechanical property of the nanowire are assume to be same and isotropic. E and ν are Young's modulus and Poisson's ratio, respectively. The inclusion is assumed to subjected to the isotropic dilatational eigenstrain ε . The functions $P(l_A/a)$, $Q(l_B/a)$, $S(l_A/a, l_B/a, H)$, $T(l_A/a, l_B/a, H)$ are given by

$$P \left(\frac{l_A}{a} \right) = 1 + \int_0^\infty \frac{G(t)}{l_A} \sin^2 \left(\frac{t l_A}{2a} \right) dt \quad (8)$$

$$Q \left(\frac{l_B}{a} \right) = 1 + \int_0^\infty \frac{G(t)}{l_B} \sin^2 \left(\frac{t l_B}{2a} \right) dt \quad (9)$$

$$S \left(\frac{l_A}{a}, \frac{l_B}{a}, H \right) = \int_0^\infty \frac{G(t)}{2l_A} \sin \left(\frac{t l_B}{2a} \right) \sin \left(\frac{t l_B}{2a} + \frac{t H}{a} + \frac{t l_A}{a} \right) \cdot \sin \left(\frac{t l_B}{2a} + \frac{t H}{a} \right) dt \quad (10)$$

$$T \left(\frac{l_A}{a}, \frac{l_B}{a}, H \right) = \int_0^\infty \frac{G(t)}{2l_B} \sin \left(\frac{t l_A}{2a} \right) \sin \left(\frac{t l_A}{2a} + \frac{t H}{a} + \frac{t l_B}{a} \right) \cdot \sin \left(\frac{t l_A}{2a} + \frac{t H}{a} \right) dt \quad (11)$$

Here, the function $G(t)$ is written as

$$G(t) = \frac{8a(1+\nu)}{\pi t^3} \left[-t \left(\frac{I_0(t)}{I_1(t)} \right)^2 + t + \frac{2}{t} (1-\nu) \right]^{-1} \quad (12)$$

where $I_0(t)$ and $I_1(t)$ are the modified Bessel functions of the zeroth order and the first order, respectively. Due to the complex interaction between the two segments, the elastic energy is difficult

to explicitly expressed by the spacing, which involves complex function. We can plot the varying of W with the changing of spacing (H) in Figure R3.

Figure R3. (a) Schematic diagram of a cylindrical rod with two inclusions. (b) Variations of the dimensionless energy density W^* with H/a .

4. It is very nice to see the evolution of the segmentation in the NW in the experiments (Fig. 1g). Is it possible for the authors to come up with an expression for the time dependence of the spacing between the segments, guided by the model and scaling based theoretical analysis. The time exponent of the size and spacing between the phases would be a very nice result, analogous to Ostwald ripening in the presence of an external flux (although I admit that the analogy is far from perfect).

** Thanks for the helpful comments. The stress driven ordering process is quite different from the Ostwald ripening process. During the Ostwald ripening process, the character length of the inclusion varies monotonously. However, during the ordering stage, the spacing between adjacent segments may decrease due to the attracting mechanism, or increase due to the repelling mechanism, even disappear as a result of the adjacent segments merging. So it is difficult to find an expression to depict the spacing varying with time, which depends on the initial condition of the two adjacent inclusion. The main reason is that the spacing is determined by the elastic energy, while the relationship between the elastic energy and the spacing is complex. How to find an appropriate expression for the spacing may be our future work.

5. Under what parameters do the authors see a transition to the Kirkendall effect? Can they see this transition in the phase-field model? If so, this would be a useful guide for the experiments. In general, the flavor of the article appears to be explaining the experimental observations. I would like the authors to show that the model is predictive, that is can it predict the segmentation in a new system which is then validated by the experiments? I realize that the authors have explored many systems, and perhaps this is simply a matter of reorganizing the presentation of the results.

** Thanks for the professional questions and valuable suggestions. The Kirkendall effect in nanomaterial synthesis is described as “pores form because of the difference in diffusion rates between two components in a diffusion couple” in the exciting work of Alivisatos’s in 2004 (*Science* **2004**, 304, 711–714). In this work, the Kirkendall effect is utilized to transform the solid Te segment in Te/Ag₂Te into hollow Pt segment. And the formation mechanism of hollow Pt segment was illustrated in our previous work (*Adv. Mater.* **2009**, 21, 1850–1854). At the early stage, some void Pt nanoshell generated *via* the galvanic replacement reaction between PtCl₆²⁻ and Te ($PtCl_6^{2-} + Te + 3H_2O \rightarrow Pt + TeO_3^{2-} + 6Cl^- + 6H^+$). The PtCl₆²⁻ and TeO₃²⁻ diffused across Pt nanoshell in the opposite directions. Therefore, the Pt nanoshell grew inward while the Te template inside was

consumed at the equivalent molar amount. Due to the molar volume of Pt ($9 \text{ cm}^3 \text{ mol}^{-1}$) is smaller than that of Te ($20 \text{ cm}^3 \text{ mol}^{-1}$), the hollow tubular Pt segment was formed.

Note that the Te/Ag₂Te nanowire is a phase-separated system, involving the migration of Ag within the fixed lattice formed by Te. The mutual diffusion Kirkendall effect may play a marginal role in this system because there only exist one kind of mobile element. Although phase field model proposed by other researchers can simulate the Kirkendall effect, the phase field model proposed in this article does not consider this effect. This model exhibits a degree of predictive capacity, including the anticipation of inclusion morphology within nanowires and its dynamic formation processes. These predictions are contingent upon the prior acquisition of specific material parameters, including mechanical properties (Young' modulus, mismatch strain and so on) and dynamic properties (diffusion coefficient). If certain key parameters are missing, the model can be used for parameterized study.

Minor comments:

1. I think the title does not do justice to the article, as it is a bit vague and does not communicate the focus of the article.

** Thanks for the comments. Based on the referee's kind suggestion, we have adjusted the title to "Ordering evolution and post-transformation enabled a library of one-dimensional segmented heteronanostructures", specifying the underlying driving force of the evolution of segmented nanostructures.

2. Perhaps a better phrase for near-attracts, far-repels? If it is used heavily in prior literature, then it's okay.

** Thanks for the suggestion. Based on the referee's kind suggestion, we have replaced it with "stress induced ordering" to accurately express the ordering process of segmented heterostructures and also revised the new version of manuscript.

3. Some words do not fit in well with the sentences (*e.g.* "vindicated"). I would advice the authors to simplify the language to so that the intent is conveyed as they intended.

** Thanks for the helpful comments and advice. The word "vindicated" is wrongly used in the manuscript, which is replaced with "confirmed" in "The biphasic attribute is further vindicated by the XRD pattern...". And the language has been revised carefully in the whole manuscript. For example, "To probe into the growth processes of Te/Ag₂Te SHs, we captured their morphologies ..." is modified as "To probe into the growth process of Te/Ag₂Te SHs, we captured their morphologies ...". And "Due to the high reactivity of Te NWs and the quick reaction with Ag⁺" is simplified into "Due to the high reactivity of Te NWs with Ag⁺". Please see the other changes in the new version of manuscript.

Reviewer: 2

This work highlights an interesting approach to synthesize segmented nanowire heterostructures starting from Te nanowires. The formation of the Te/Ag₂Te segmented heterostructures is discussed

in great detail, also with *in situ* microscopy techniques and is additionally modeled. From these heterostructures, other heterostructures can be generated in a two-step approach. Various groups have previously covered the formation of heterostructures but the mechanism reported in this work appears new. I have the following remarks:

** Thanks for the valuable and nice comments. In this work, we explored the driving force for the formation of 1D segmented heterostructures using Te/Ag₂Te as a prototype. By capturing the morphologies of different evolution stages of Te/Ag₂Te combined with the phase field model, we proposed the three-stage evolution mechanism for segmented heterostructures, that is defect-assisted island generation, stress-induced strip formation and elastic energy-driven ordering. With the simple post-transformation processes using the segmented heterostructures as templates, more segmented heterostructures were easily prepared.

1. The whole story hinges on Te nanowires and is therefore doubtful that the findings of the work can be easily extended to other materials.

** Thanks for the comments. Te/Ag₂Te was selected as a model and their formation process is explained in detail. In addition to the Te/Ag₂Te system, Te/PbTe and Te/Cu_{1.75}Te systems were also explored in New Figure S23. Similarly, with the under-stoichiometric synthesis strategy, Te/PbTe and Te/Cu_{1.75}Te segmented nanowires were successfully prepared and their formation processes were further simulated by the phase field model. The calculation results are shown as bellows, which is also in agreement with the experiments. Factually, Te nanowires are selected as a template model based on the following considerations. Their uniform morphology and single component are suitable for the ideal model for both physical and chemical templates in reactions. Their high reaction activity with metal ions is conducive to the subsequent chemical transformation to enrich the nanomaterial library. And their easily scalable synthesis is also promising for the practical application in the near future. Besides, telluride materials offer significant advantages, such as high theoretical volume capacity and electrical conductivity, showing promising prospects for different energy-related applications.

New Figure S23. Simulated results of Te/PbTe and Te/Cu_{1.75}Te SHs. a, b, The calculated dimensionless elastic energy versus the differences in length between adjacent segments in Te/PbTe SHs, segment length (b) and segment separation (c). c, d, The calculated dimensionless elastic energy versus the differences in length between adjacent segments in Te/Cu_{1.75}Te SHs, segment length (c) and segment separation (d).

2. The mechanism proposed here should have a strong dependence on the diameter of the wires, but this is a critical parameter that the authors are not exploring. To really prove the mechanism, which should most likely predict different sizes of the segments based on strain balance, depending on wire diameter, it is advisable indeed to experimentally probe wires with different diameters. If this is not done, in my view the overall explanation and model are somewhat speculative.

** Thanks for the valuable comments and helpful advices. We agree that the radius of the nanowire is an important factor that affects the elastic energy dominated near-attraction far-repulsion mechanism. How the nanowire radius affects the ordering driving force, *i.e.*, the elastic energy, has been shown in the New Figure 3e.

From the viewpoint of driving force, the near-attracts far-repels paradigm of two adjacent cylinder inclusion in a nanowire determined by the elastic interaction. For the given segment length, as illustrated in New Figure 3e, when the spacing between two segments is less than the critical value, decreasing the spacing between them results in a reduction in the system's elastic energy. On the other hand, when the spacing between two segments exceeds the critical value, an increase in spacing also leads to a reduction in the system's elastic energy. It should be pointed out that the radius of the nanowire is not an independent factor that affects the driving force for the interaction of two adjacent segments, it is the ratio of segment spacing to the radius.

New Figure 3e. The calculated dimensionless elastic energy versus ratios of segment separation to the radius (H/a).

As an example, for the given segment length ($h=30$ nm) and the spacing between two segments ($H=20$ nm), we simulate the interaction process between two segments in the Ag/Te nanowire with the radius of 25 nm and 30 nm, respectively. The morphology of Ag_2Te segments and the corresponding distribution of dimensionless hydrostatic stress at different dimensionless times, ($t^*=0, 200, 400, 600, 800$) are shown in Figures R4 and R5. In both cases, the ratios $H/a=0.8$ and $H/a=0.67$ are smaller than the critical value. As a result, the adjacent segments merge in these two nanowires.

Figure R4. The simulation of the interaction process between two segments with the initial spacing $H=20$ nm in the nanowire with the radius of 25 nm at different dimensionless times, $t^*=0, 200, 400, 600, 800$. (a)-(d) The morphology of Ag_2Te segments. (e)-(h) The distribution of dimensionless hydrostatic stress corresponding to (a)-(d).

Figure R5. The simulation of the interaction process between two segments with the initial spacing $H=20$ nm in the nanowire with the radius of 30 nm at different dimensionless times $t^*=0, 200, 400, 600, 800$. (a)-(d) The morphology of Ag_2Te segments. (e)-(h) The distribution of dimensionless hydrostatic stress corresponding to (a)-(d).

Similarly, for the given segment length ($h=30$ nm), the spacing between two segments $H=30$ nm in the $\text{Ag}_2\text{Te}/\text{Te}$ nanowire with the radius of 25 nm and the spacing between two segments $H=36$ nm in the $\text{Ag}_2\text{Te}/\text{Te}$ nanowire with the radius of 30 nm, we simulate the morphology of Ag_2Te segments and the corresponding distribution of dimensionless hydrostatic stress at different dimensionless times, ($t^*=0, 200, 400, 600, 800$), which are shown in Figures R6 and R7, respectively. In both cases, the ratios $H/a=1.2$, which are larger than the critical value. As a result, the adjacent segments are exclusive in these two nanowires.

Figure R6. The simulation of the interaction process between two segments with the initial spacing $H=30$ nm in the nanowire with the radius of 25 nm at different dimensionless times $t^*=0, 200, 400, 600, 800$. (a)-(d) The morphology of Ag_2Te segments. (a)-(d) The morphology of Ag_2Te segments. (e)-(h) The distribution of dimensionless hydrostatic stress corresponding to (a)-(d).

Figure R7. The simulation of the interaction process between two segments with the initial spacing $H=36$ nm in the nanowire with the radius of 30 nm at different dimensionless times $t^*=0, 200, 400, 600, 800$. (a)-(d) The morphology of Ag_2Te segments. (a)-(d) The morphology of Ag_2Te segments. (e)-(h) The distribution of dimensionless hydrostatic stress corresponding to (a)-(d).

Figure R8. TEM images of $\text{Te}/\text{Ag}_2\text{Te}$ using thicker Te NWs as templates.

During the fabrication of the Te nanowire, when the radius of the nanowire increases, the length of the nanowire will decrease. As a result, many nanowire ends appear in the experimental observation, as shown in Figure R8. Further, we investigate how the nanowire end affects the formation of the Ag_2Te segment. In Figures R9 and R10, we simulate the formation of the Ag_2Te segments in the end of a nanowire with the radius of 15 and 19 nm, respectively. We can see that the Ag_2Te segment prefers to form in the end of the nanowire, which is consistent with the experimental observation.

Figure R9. Simulation of the Ag_2Te segments formation in the end of a nanowire with the radius of 15 nm. (a)-(f) The evolution of Ag_2Te inclusion morphology at different times $t^*=100, 500, 900, 1300, 1700, 2100$.

Figure R10. Simulation of the Ag_2Te segments formation in the end of a nanowire with the radius of 19 nm. (a)-(f) The evolution of Ag_2Te inclusion morphology at different times $t^*=100, 500, 900, 1300, 1700, 2100$.

3. The second part of the work is considerably less interesting, as the heterostructures are then transformed into other heterostructures with different compositions following well-known procedures (such as cation exchange).

** Thanks for the comments. The post-transformation of segmented heterostructures into other heterostructures are designed to enrich the library of nanomaterials. Extensive post-transformation, such as cation exchange, galvanic reaction, Kirkendall effect and other existing simple procedures, also imply that these segmented structures prepared here have great potential to be transformed into more complex and functionalized nanomaterials taking into account the advantages of heterogeneous structures and interfaces. These independent post-transformation processes, which can also be programmed, also indicate that these segmented structures have good structural stability and controllability. In addition, expanded palette of nanomaterials always powers the development of nanoscience and evolution of society. Regular pattern design in 1D segmented heterostructures are fundamentally impressive due to the synergistic effect between different components and interfaces. Controlled position of various materials within one nanostructure is significantly important for the function integration. Heterogeneous interfaces determine the electronic and magnetic coupling between multiple compositions. Therefore, the rational design and precise synthesis of 1D segmented heterostructures plays a decisive role in the applications of next-generation nanostructures across many fields.

Currently, only limited 1D heterostructured nanomaterials, such as sulfides and selenides, have been prepared. Telluride materials offer significant advantages in terms of their high theoretical volume capacity and high electrical conductivity, making them promising candidates for different energy-related applications. Therefore, a simple and scalable platform for constructing the 1D telluride segmented heterostructures library has become a crucial issue to resolve. This work makes up for the lack of telluride heterostructures and facilitates their widespread applications.

4. In this work the TEM images cover limited areas of each sample. Being TEM a very local technique, one can always scan the grid in a region showing samples that appear to confirm one's hypothesis. Hence for each of the transformations reported here the authors should present a larger set of TEM images, including wide fields of view, supporting their claims.

** Thanks for the comments. Based on the referee's kind suggestion, we have taken larger sets of TEM images of samples to show the morphologies more clearly. Some examples are shown as follows. Please see the new version of manuscript for more revisions. As shown in Figures S1a, S15a and R11, the large-scaled TEM images of Te/Ag₂Te, Te/PbTe, and Te/Bi₂Te₃ segmented heterostructures demonstrate the uniformity of segmented structures.

New Figure S1a. Large-scale TEM image of Te/Ag₂Te SHs.

New Figure S15a. Large-scale TEM image of Te/PbTe SHs.

Figure R11. Large-scaled TEM image of Te/Bi₂Te₃ segmented SHs.

Reviewer: 3

Overall, this manuscript describes interesting synthesis and characterization studies of onedimensional (1D) heteronanostructures, including nanowire-nanowire (NW-NW) and nanowire-nanotube (NW-NT) materials. Over the past several decades there has been considerable interest in 1D heteronanostructures, often being termed axial or radial superlattices since previous work generally focused on semiconductor-semiconductor and/or semiconductor-metal structures originally studied in the context of planar (2-dimensions, 2D) superlattices. The current focus on a broad range of metal heteronanostructures breaks new ground and can make a unique and important contribution to the literature that this reviewer believes will be

interesting to readers of *Nature Comm.* as well as the active field of nanostructure synthesis. Specific strengths and weaknesses of the work are as follows.

** Thanks for the valuable and nice comments. The research on 1D metal heteronanostructures represents a significant development in the field of nanostructure synthesis. While previous studies mainly focused on 2D semiconductor-semiconductor and/or semiconductor-metal structures, this work explores 1D metal heteronanostructures, which will open up new possibilities in nanomaterial applications. Currently, a general synthesis method for 1D ingenious and intricate nanostructures still remains formidable. In this work, a simple and versatile approach based on the under-stoichiometric reaction strategy is innovatively proposed to synthesize 1D segmented heterostructures. We anticipate that this work would open a new avenue in 1D nanomaterial diversity field and cast light on the understanding the formation of 1D periodic heterostructures.

1. The general approach described in the manuscript is quite interesting and compliments previous work. For example, there has been considerable effort (related to 2D superlattices) whereby sequential delivery of reactants has been used to prepare a broad spectrum of axial and radial NW superlattices. Also, phase separation strategies trading off thermodynamic and kinetic constraints have also led to heteronanostructures superlattices. The new work might be claimed to be most related to this later area of prior work, but I personally feel, the concept and approach remain quite distinct and original; that is, it should stimulate considerable interest by other researchers to apply this to other materials as well.

** Thanks for the nice comments. Expanded palette of nanomaterials always powers the development of nanoscience and evolution of society. 1D nanomaterials have inspired considerable interest from researchers due to their exotic physical and chemical properties, which results from their diameters falling into the wavelength of light, the mean free path of phonons, the exciton Bohr radius, the critical size of magnetic domains, and the exciton diffusion length. 1D axial heterostructures intrinsically show modulable spectral and transmissive properties derived from their symmetry and periodicity, as well as the combination of various components and interfaces. This work is aimed to provide a broad tool set for the synthesis of 1D axially segmented heterostructures, expanding the palette of material selection. We anticipate that this work would complement previous work in the field of 2D superlattices and stimulate considerable interest and further advancements in different material system.

2. The TEM data—static images and movies—provide strong evidence supporting the proposed model and successful synthetic realization of the wide-range of heteronanostructures (*i.e.*, Fig. 1, Fig. 4a, supplementary movies). Nevertheless, this data also might be one of the weaker points of the paper, especially within the context of model structures/modeling shown, for example, in Figs. 2 & 3. The hallmark of previous work in the field has been very clear images of the axial and radial superlattice repeats and material boundaries/interfaces often with high-resolution atomic images. While I do not believe the present level of TEM data should disqualify the paper, I would also urge the authors to see whether they can provide sharper data, including movies, that illustrate better 1-2 of heteronanostructures materials.

** Thanks for the helpful comments. Based on the referee's kind suggestion, we have taken more clear TEM images of samples to show the segmented morphologies. In addition to the morphological images, we also performed some spectral characterizations. Some examples are

shown as below.

New Figure S2. Morphological and structural characterizations of Te NWs, Ag₂Te NWs and Te/Ag₂Te SHs. a, b, c, TEM images of Te, Ag₂Te, and Te/Ag₂Te. d, HRTEM images of Te/Ag₂Te SHs, with spacings of 0.197 and 0.283 nm corresponding to Te and Ag₂Te, respectively. Inset: HRTEM image of Te/Ag₂Te SHs. This observation potently establishes that the heterogeneous NW contained Te and Ag₂Te phase alternations along its axes. e, XRD pattern analysis of Te/Ag₂Te SHs. f, The survey XPS spectra, showing the existing Ag element in Te/Ag₂Te. g, Fitted XPS spectra of Te 3d orbital, showing the presence of both Te⁰ and Te²⁻ phases. h, Raman spectra of Te/Ag₂Te SHs.

New Figure S7. Detailed characterization of initial islands. a, HADDF image and EDS mapping of Te and Ag. b, c, HADDF images of islands on NW.

New Figure S20. Morphological and structural characterizations of Te/Bi₂Te₃ SHs. a, HADDF-STEM and elemental mapping characterizations. b, c, HRTEM images with spacings of 0.306 and 0.218 nm corresponding to Te and Bi₂Te₃, respectively. d, EDS.

New Figure S21. Morphological and structural characterizations of Te/Bi₂Te₃ SHs with various ratios of Bi/Te. a, b, c, d, TEM images with Bi/Te=10, 5, 2.5 and 1.25. e, f, XRD patterns of Te/Bi₂Te₃ with different quantities of Bi precursor and NaOH. The peak at 23.2° attributed to Te

weakens with the increasing Bi source, while the peak at 40.9° attributed to Bi_2Te_3 strengthens with the increasing NaOH.

New Figure S24. Morphological and structural characterizations of Te/ZnTe SHs. a, b, TEM and HRTEM images, showing the spacings of 0.38 and 0.23 nm corresponding to Te and ZnTe, respectively. c, XRD pattern. d, HADDF-STEM and elemental mapping characterizations. e, EDS. f, Line mapping profiles.

3. It would be better in this reviewer's opinion not to somewhat bury the thermoelectric property characterization in Supplementary Figure 44. We agree with the authors that it is important at this stage in the field to demonstrate a potential direction that is enabled by a new nanomaterial, so the effort to characterize thermoelectric properties is appreciated. Nevertheless, I think the paper could be strengthened by addressing this key result (it is in fact a motivation for others why they should be interested in the work if not simply interested in materials synthesis) in considerably more depth. Specific points that the authors should consider might include the following:

(a) perhaps it would be more effective to include a clear description of the thermoelectric measurements and results in a figure in the main text (e.g., the last main text figure);

** Thanks for the professional comments and valuable suggestions. We have added the thermoelectric property characterization in the main text with revised Figure 5.

New Figure 5. The thermoelectric performance and band structure of Ag₂Te/PbTe 1D SHs. (A) Seebeck coefficient, (B) power factor of Ag₂Te/PbTe 1D SHs compared with PbTe NWs, Ag₂Te NWs, and Ag₂Te+PbTe NW mixture between 425 and 775 K. (C) Band structure in PbTe NWs and Ag₂Te NWs before contact. (D) Equilibrium band alignment in Ag₂Te/PbTe 1D SHs.

(b) it is unclear how the measurements were carried out and this is important to the interpretation (and possible criticism by experts in this active research area). For example, ideally it the most meaningful measurement would be made on *individual* heteronanostructures; yes, they are most challenging but have been achieved by several groups in the field. If the measurements were, however, carried out on ensembles of the heteronanostructures it is unclear whether results reflect the intrinsic properties of the heteronanostructures, the differing nanostructure-to-nanostructure electrical/thermal transport (which would vary with materials due to surface differences) or some combination of the two.

** Thanks for the comments. The thermoelectric property measurement was conducted on the ensembles of heteronanostructures. The ensembles were prepared as follows: Firstly, the surfactant was removed *via* stirring for 3h in diluted hydrazine hydrate solution of ethanol (10% volume ratio) as reported previously (*Nano Lett.* **2012**, 12, 1, 56–60). The precipitant was washed and then dried in vacuum for further pressing. The pressed cylindrical pellet for basic thermoelectric measurement has a diameter of 12.7 mm and a height of 2 mm. The Seebeck coefficient and electrical conductivity were measured under a low-pressure He atmosphere from 425 to 775 K with a CTA instrument (Cryoall, China). The Power factor is calculated as: $PF = \frac{S^2 \times \sigma}{K}$.

4. Probably least important to this reviewer but a topic that can cause anger is the citations to the literature. I think the current version of the manuscript has a somewhat random choice of citations (this is made difficult by the vastness of the literature), and I would like to respectfully suggest the following to the authors. Cite several of the most early (seminal) NW-NW and NW-NT reports, but also cite reviews from the major groups, including at least one recent review, that have at least partially focused on this subject because these reviews have generally cited comprehensively many more papers than can be cited in an original research publication (this can help to allay disagreement with authors who feel their work should have been cited).

** Thanks for the professional comments and helpful suggestions. Based on the referee's kind remind, we have carefully revised the whole citations added the following citations at the suitable place in the new version of manuscript. These references include the pioneering works from Charles M. Lieber, Paul Alivisatos, Raymond E. Schaak, Peidong Yang, Chad A. Mirkin, William E. Buhro, R. S. Wagner, and Yadong Yin *et. al* about the cation exchange method, vapor-liquid-solid growth method, solution-liquid-solid growth method *ect.* in the synthesis of 1D heterostructures. And the latest review publications about the preparation of heterostructured nanomaterials are also contained. Besides, the recent work and summary of our research group on 1D heterostructures are also cited.

1. A laser ablation method for the synthesis of crystalline semiconductor nanowires, *Science* **1998**, 279, 208–211.
2. Growth of nanowire superlattice structures for nanoscale photonics and electronics, *Nature* **2002**, 415, 617–620.
3. Flow-based solution–liquid–solid nanowire synthesis, *Nat. Nanotech.* **2013**, 8, 660–666.
4. Nanowired bioelectric interfaces, *Chem. Rev.* **2019**, 119, 9136–9152.
5. Cation exchange reactions in ionic nanocrystals. *Science* **2004**, 306, (5698), 1009–1012.
6. Colloidal nanocrystal heterostructures with linear and branched topology, *Nature* **2004**, 430, 190–195.
7. Colloidal nanocrystal synthesis and the organic-inorganic interface. *Nature* **2005**, 437, (7059), 664–670.
8. Spontaneous superlattice formation in nanorods through partial cation exchange, *Science* **2007**, 317 (5836), 355–358.
9. Vapor phase growth of semiconductor nanowires: key developments and open questions, *Chem. Rev.* **2019**, 119 (15), 8958–8971.
10. Combinatorial cation exchange for the discovery and rational synthesis of heterostructured nanorods, *Nat. Synth.* **2023**, 2, 152–161.
11. Rational construction of a scalable heterostructured nanorod megalibrary, *Science* **2020**, 367 (6476), 418–424.
12. Made-to-order heterostructured nanoparticle libraries, *Acc. Chem. Res.* **2020**, 53, 11, 2558–2568.
13. Tunable intraparticle frameworks for creating complex heterostructured nanoparticle libraries, *Science* **2018**, 360 (6388), 513–517.
14. Block-by-block growth of single-crystalline Si/SiGe superlattice nanowires, *Nano Letters* **2002**, 2 (2), 83–86.
15. Interface and heterostructure design in polyelemental nanoparticles, *Science* **2019**, 363, (6433), 959–964.
16. Solution-liquid-solid growth of crystalline III-V semiconductors: an analogy to vapor-liquid-solid growth. *Science* 1995, 270(5243), 1791–1794.
17. Vapor-liquid-solid mechanism of single crystal growth. *Appl. Phys. Lett.* **1964**, 4(5), 89–90.
18. Engineered interfaces for heterostructured intermetallic nanomaterials, *Nat. Synth.* **2023**, 2, 749–756.
19. Axially segmented semiconductor heteronanowires, *Acc. Mater. Res.* **2020**, 1(2), 126–136.
20. One-dimensional superlattice heterostructure library, *J. Am. Chem. Soc.* **2021**, 143(18), 7013–7020.
21. Pulsed axial epitaxy of colloidal quantum dots in nanowires enables facet-selective passivation,

Reviewer: 4

The authors synthesize Ag₂Te/Te nanowires with striped periodicity and attribute this to a “near attracts far repels” hypothesis that comes out of phase field modeling. They then apply a range of established chemical transformation reactions to convert this to a library of derivative multi-component nanowires. There are some potentially interesting hypotheses and results here, but I do not recommend the manuscript for publication in *Nature Comm*. I find inconsistencies between the experimental data and the model and I struggle to link what is claimed as “near attracts far repels” to what is shown experimentally. This was also a very confusing paper to read and I fear that even experts in the field will struggle to comprehend it. I explain more about my rationale below.

** Thanks for the comments. We have enriched the experimental data of this work, especially the morphological characterization part. The “near attracts far repels” is replaced with “stress induced ordering” to more clearly express the ordering process of segmented heterostructures. And the language in new version of manuscript is also simplified and modified so that the intent conveyed is easier to understand. We think that the reviewer may misunderstand the mechanism proposed in this paper. The main reason may be attributed to the complex process, which is separated into two main stages (the growth stage and the ordering stage), the near attracts far repels mechanism dominates in the ordering stage. The reviewer may misunderstand when the reviewer think the ripening stage still using the near attracts far repels mechanism. We should emphasize the near attracts far repels mechanism play the role in the ordering stage to avoid the misunderstanding. We further improve the quality of the manuscript to make it more clear to the readers.

1. Introduction: Some important aspects of the field are highly over-sold and over-hyped. At the same time, other aspects are unrealistically over-simplified. It would be appreciated if the authors could better balance this.

** Thanks for the valuable comments. We again review the field of solution-phase synthesis of 1D heterogeneous nanostructures and make an important revision of the Introduction part of manuscript. We first summarized the unique advantages of 1D axial heterostructures derived from their multi-component and interface. The controlled position of various materials within one nanostructure is significantly important for the synergistic effect between different components. The interfaces connecting multiple components determine the electronic and magnetic coupling. Therefore, the rational design and precise synthesis of 1D heterostructures plays a decisive role in the applications of next-generation nanostructures across many fields and the importance of precisely engineering 1D heterostructures is emphasized. Then, we state the fact that only limited 1D heterostructured nanomaterials, such as sulfides and selenides, have been prepared. Telluride materials offer various advantages, such as high theoretical volume capacity and electrical conductivity, showing promising prospects for different energy-related applications. Unfortunately, the synthesis of 1D telluride heterostructures has been rarely reported. Next, we recapitulate our motivation for preparing these 1D telluride heterostructures. We aim to develop a simple and scalable platform for constructing 1D telluride heterostructure library, as well as exploring their evolution mechanism. The phase field method is employed in view of its ability of modeling and predicting the mesoscale morphological and microstructure evolution in materials, especially outstanding in describing the phase

transformation coupled with multi-field physics. Therefore, we describe our strategy to obtain these 1D telluride heterostructures and their formation mechanism *via* phase field model. Finally, we outlook the significance of this work in the field of nanomaterial synthesis. Please see the revised *Introduction* part in the new version of manuscript.

2. Precedent: Most of what is described for the Ag^+ reaction with Te nanowires, forming Te/ Ag_2Te , seems akin to the initial Science paper on $\text{Ag}_2\text{Se}/\text{CdSe}$ superlattice nanowires by Alivisatos, in terms of the (hypothesized at the time) process and the heterostructured morphology. In the Alivisatos paper, this very similar ordering and bulging was attributed (through computational modeling) to strain. How do the authors reconcile this difference in proposed mechanism?

** Thanks for the professional comments. We agree that the morphology of $\text{Ag}_2\text{Te}/\text{Te}$ superlattice nanowire is similar with the $\text{Ag}_2\text{Se}/\text{CdSe}$ superlattice nanowires in the Science paper by Alivisatos. Both the Science paper and our paper analyze how the elastic energy affect the superlattice morphology. It should be noted that our work is different from the Science paper in the following three aspects.

(1) The *Science* paper published by the Alivisatos group conducted a analysis about the role of the elastic energy by comparing the magnitude of strain energy under different morphological conditions for the with cylinder inclusion the fixed length, aiming to ascertain the repulsive effects of strain energy on adjacent inclusions. Note that the cylinder inclusion length is fixed in their calculation. It should be pointed out that, this paper did not offer an explanation for why inclusions are of equal length in superlattice structures and why their spacing is also equidistant. This important problem had been solved in our paper. Please see Figure R1, we demonstrate that the dimensionless elastic energy (W^*) of the system increases monotonically with increasing the discrepancy of the length between adjacent Ag_2Te ($\Delta h = h_1 - h_2$) and the Te segments ($\Delta H = H_1 - H_2$), respectively. The elastic energy reaches a minimum value at ($\Delta h = 0$) and ($\Delta H = 0$).

New Figure 3d. The calculated dimensionless elastic energy versus the differences in length between adjacent segments

(2) The *Science* paper explain the ordering phenomenon, *i.e.*, the repulsion of the adjacent cylinder inclusion, from the perspective of thermodynamics. It should be pointed out that the *Science* paper considered the diffusion effect only in the Ostwald ripening process, *i.e.*, the island growing stage, not the segment ordering stage. Furthermore, our work simulates and analyze the formation process of the superlattice structure from the kinetic and thermodynamic viewpoint, respectively, especially in the ordering stage. From the thermodynamic viewpoint, we analyzed that elastic driving force for the near-attraction far-repulsion mechanism. Through the kinetic analysis, the import role of the diffusion coefficient of the guest atom in the host nanowire is investigated. As an example, when the diffusion coefficient of the guest atom decreases from 5.0×10^{-10} to $5.0 \times 10^{-13} \text{ cm}^2 \text{ s}^{-1}$, the

coalesce of the adjacent segments will not occur (see Figs. R2 and R3), which can not be provided by thermodynamic analysis.

Figure R12. The simulation of the interaction process between two segments with the diffusion coefficient $5.0 \times 10^{-10} \text{ cm}^2 \text{ s}^{-1}$ at different dimensionless times ($t^*=0, 400, 800, 1200$). (a)-(d) The morphology of Ag_2Te segments. (e)-(h) The distribution of dimensionless hydrostatic stress corresponding to (a)-(d).

Figure R13. The simulation of the interaction process between two segments with the diffusion coefficient $5.0 \times 10^{-13} \text{ cm}^2 \text{ s}^{-1}$ at different dimensionless times ($t^*=0, 400, 800, 1200$). (a)-(d) The morphology of Ag_2Te segments. (e)-(h) The distribution of dimensionless hydrostatic stress corresponding to (a)-(d).

(3) Our phase field model can investigate the influence of anisotropic interfacial energy on the inclusion's configuration. To clarify the effect of interfacial energy, we first omit the contribution of elastic energy by setting the elastic energy density to be 0. When the ratio κ_x/κ_y is set as 1/4, the simulation results of the growth of Ag_2Te island are shown in New Figure S10a. We can see that the growing Ag_2Te island yields segment in the nanowire, but the island morphology during growth is not consistent with the experiential observation in the original manuscript. Notably, when the ratio is set as $\kappa_x/\kappa_y=4$, the simulation results of the growth of Ag_2Te island are shown in New Figure S10b. In this case, the Ag_2Te island even dose not yield segment in nanowire due to the large ratio of κ_x/κ_y .

New Figure S10. Effect of anisotropic interfacial energy on the growth of Ag_2Te island without considering the stress at different dimensionless times ($t^*=200, 400, 600, 800$). a, The anisotropic interfacial energy parameter $\kappa_x/\kappa_y=1/4$. b, The anisotropic interfacial energy parameter $\kappa_x/\kappa_y=4$.

When the expansion coefficient is set as 0.04, the Ag_2Te inclusion growth process is also

simulated as the interfacial energy parameter $\kappa_x/\kappa_y=1/4$ and $\kappa_x/\kappa_y=4$ in New Figure S11, respectively. We can see that the morphology of Ag_2Te inclusion is similar with the experimental observation. It can be concluded that during the dynamic process of the island inclusion growth into cylindrical configuration, the stress plays a much more important role than interfacial energy, although interfacial energy has an influence on the dynamics.

New Figure S11. Effects of anisotropic interfacial energy and mismatch strain on the growth of Ag_2Te island. a, b, The morphology of Ag_2Te island at different dimensionless times ($t^*=200, 400, 600, 1000$) and the corresponding distribution of dimensionless hydrostatic stress when the anisotropic interfacial energy parameter κ_x/κ_y is set as $1/4$, and the expansion coefficient is set as 0.04 . c, d, The morphology of Ag_2Te island at different dimensionless times ($t^*=200, 500, 800, 1200$) and the corresponding distribution of dimensionless hydrostatic stress when the anisotropic interfacial energy parameter κ_x/κ_y is set as 4 , and the expansion coefficient is set as 0.04 .

(4) In the ordering stage, based on the kinetic modeling, we propose and prove the mechanism of stress driven merging of adjacent cylinder inclusions. Effect of mismatch strain on the interaction process between two segments is revealed in our work. Note that the migration of ions requires to overcome energy barriers between two positions. If the mismatch strain is quite small, the driving force will be not enough to drive the spacing change of the inclusion. If the mismatch strain is large enough to overcome the energy barrier of the guest atom migration, the ordering process will occur. The large mismatch strain will facilitate the ordering. As an example, we investigate the interaction between two segments with the initial space $H/a=1$, with the mismatch strain increasing from 0.005 to 0.045 , the morphology evolution is shown in New Figure S22. As expected, for the small mismatch strain (0.005), the space between two segments hardly changes during evolution, because of the small elastic driving force. When the mismatch strain increases, the segments will merge if the initial space is smaller than the critical value. The large mismatch strain facilitates the coalesce of segments due to the large driving force.

New Figure S22. Effect of mismatch strain on the interaction process between two segments at different dimensionless times ($t^*=0, 400, 800, 1200$). a, b, The morphology of Ag_2Te island and the corresponding distribution of dimensionless hydrostatic stress when the mismatch strain is set as 0.005. c, d, The morphology of Ag_2Te island and the corresponding distribution of dimensionless hydrostatic stress when the mismatch strain is set as 0.025. e, f, The morphology of Ag_2Te island and the corresponding distribution of dimensionless hydrostatic stress when the mismatch strain is set as 0.045.

3. The model: As I understand it, in this current submission, the authors create a model based on elastic energy, identifying a critical threshold for repulsive vs attractive interactions of the similar segments based on elastic energy (page 8). However, their preceding discussion (page 7) focused on diffusion and migration, yet I do not see (or understand) how diffusion of Ag^+ through Te (and aspects of structure changes and/or redox chemistry that will be important mechanistically) relates to elastic energy. So unless I am missing something, I fail to see how the experimental data relates to the model that is supposed to connect what is observed to what can be predicted. This is the premise and sales pitch for the paper, and it leaves me confused and feeling like the two are contradictory. Perhaps they are not, but the authors were not able to convince me in the current version of the manuscript.

**** Thanks for the comments.** The reviewer misunderstood the statement in our original manuscript, we rewrite the paper to make it clear. We will respond to the reviewer's questions from the following four aspects:

(1) It is imperative to point out that, besides the elastic energy, our model considered the chemical

free energy and interfacial energy, which can be seen in the governing equation of our model.

$$\rho \frac{\partial c}{\partial t} = \frac{1}{\phi} \left[\nabla \cdot \left(M \phi \cdot \nabla \left(\frac{\partial f_c}{\partial c} - \kappa \nabla^2 c + \frac{\partial f_e}{\partial c} \right) \right) \right] + \frac{1}{\phi} |\nabla \phi| I \quad (1)$$

where f_c and f_e are the chemical free energy density and elastic energy density, respectively. The term $\kappa \nabla^2 c$ denotes the contribution of interfacial energy. By the time dependent governing equation, the phase field model can depict the kinetic processes of the inclusion growth and ordering.

(2) The analysis of repulsive vs attractive interactions of the similar segments based on elastic energy is not contradictory with discussion about diffusion and migration. In the original manuscript, we first simulated the growth of the island. During the rapid reaction stage, it is common that both long and short segments form, as shown in Fig. R4. In the following ordering stage (Fig. R5), the Ag will leave from the long cylinder inclusion, migrates into the short cylinder inclusion, which is mainly driven by the elastic energy. That's why we discuss about the Ag migration in Te nanowire. Furthermore, we conduct an analysis and demonstrate that elastic energy serves as the driving force for the ordering phenomena between adjacent segments. The elastic energy will change when the length of the cylinder inclusion or the spacing between two inclusions change. however, the interfacial energy does not change due to the constant interface area. Thus, we analyze the elastic energy in detail.

Figure R14. Simulations of the growth of Ag_2Te islands into the poorly-ordered segments at times of $t^*=50$, $t^*=150$, $t^*=250$ and $t^*=350$.

Figure R15. Evolution of disordered strips to ordered segments in the NW at times of $t^*=450$, $t^*=850$, $t^*=1050$ and $t^*=1850$.

(3) The diffusion of Ag^+ through Te relates to elastic energy. The elastic energy is a part of the driving force for the diffusion of guest atom in the host material, based on thermodynamics. This effect has been considered in the phase field mode by the equation (1).

(4) It is important to note that our phase field model is related to the experimental observation. First, the phase field model successfully simulated the phenomena observed in experiments and further revealed the underlying mechanisms. The growth of the Ag_2Te island into the cylinder inclusion and the ordering process are observed in both the experiment and our simulation data. Second, the nanowire configuration, and the mechanical, kinetic properties of the Te nanowire are adopted in the field model. Thus, we believe that the phase field model is related to the experimental data.

4. My proposed alternate explanation: In terms of their “near-attracts far-repels” theory, could that not simply be described, more simply, by minimization of interfacial energies (proxy for strain), which is already well established in the field and that has already been used to predict heterostructuring in metal chalcogenide nanorods? The “near attracts” part means that two interfaces would have to co-exist with a narrow separation, and that would be energetically unfavorable, so they combine. The “far repels” part means that two interfaces are far enough apart to not have to interact, and they remain separated and become located at ideal locations for other reasons (strain minimization?). I am not saying that my proposal is accurate, but rather that it perhaps provides a reasonable alternative to elastic energy that, to my current level of understanding, makes a bit more sense in connecting the mechanistic pieces together. In the end, I am left struggling to figure out how elastic energy is relevant. The authors would have to convince me a bit more to help connect this rationale to the details of the system and reaction.

** Thanks for the helpful comments. We think the reviewer misunderstand the mechanism, the reviewer may not notice that the range affected by the interface energy is different from that by the strain energy. Generally, the interface energy only dominates in the vicinity of the interface between the inclusions and the matrix, while the range affected by strain energy is typically much larger than that by interface energy and is associated with the characteristic dimensions of the inclusions. we rewrite this part to make it clear.

Note that different mechanisms dominate at different stages of superlattice structure formation. In the growing stage, Ag ions are inserted into the Te nanowire driven by the reaction, leading to the growth of Ag_2Te inclusions. After the reaction ceases due to the depletion of available Ag^+ ions in the solution, the Ag_2Te inclusions will reorganize, which is termed the ordering stage. In the growth stage, as island inclusions grow due to the insertion of Ag ions, when the distance between adjacent islands falls within the range influenced by interfacial energy, two islands merge. This process is primarily governed by chemical reactions and interfacial energy.

Once the Ag is no longer inserted into Te nanowire due to depletion of available Ag^+ ions in the solution, the volume of Ag_2Te will not change. For two adjacent Ag_2Te inclusion with the distance beyond the range of interfacial energy, assuming a slight translational shift in the position of an inclusion after perturbation, the interfacial area remains unchanged, as a result, both the chemical free energy and interfacial energy of the system remain unchanged. However, considering the elastic interaction between cylindrical inclusions, within a specific range of elastic energy influence, the inclusions approach each other, until they get close enough and merge together.

The new perspective introduced in the article pertains to the ordering stage and how elastic energy influences this process when interfacial energy is no longer effective. For the ‘near-attracts’ mechanism, it is important to emphasize that our analysis and the proposed strain energy-driven ordering mechanism are distinctly different from the interfacial energy-driven coarsening

mechanism.

5. Chemical transformations: The “chemical transformation” section is already established chemistry that adds materials diversity, but doesn’t relate to their “near attracts far repels” hypothesis. It’s nice and adds interest, but this much larger library of systems doesn’t seem to validate (or refute) their hypothesis as the pathway is different. (The ordering is already there before these transformations are carried out, if I understand correctly.) The actual experimental validation of the main mechanistic point of the paper is therefore quite limited.

** Thanks for the valuable comments. The post chemical transformation is designed to enrich the library of heterogeneous nanomaterials. Various post-transformation based on the already established reactions also imply that these segmented structures have great potential to be transformed into more complex and functionalized nanomaterials. These independent post-transformation processes, which can also be programmed, indicate that these segmented structures have good structural stability and controllability. Besides, only limited 1D heterogeneous nanomaterials, such as sulfides and selenides, have been prepared. Telluride materials offer significant advantages in terms of their high theoretical volume capacity and high electrical conductivity, making them promising candidates for different energy-related applications. Therefore, a simple and scalable platform for constructing the 1D telluride segmented heterostructures library has become an urgent need. This post-transformation will make up for the lack of telluride heterostructures and facilitates their widespread applications. In addition, we have also made efforts to validate the mechanism, such as their morphology being recharacterized in detail and Raman spectra being captured. The more detailed analysis of XRD pattern and XPS spectra are also made. Please see the new version of manuscript.

6. Materials characterization: In addition to the concerns mentioned above, I also have significant concerns about materials characterization. For example, in Fig 1, the XRD data does not seem to match what they claim; the authors should take a closer look at XRD interpretation throughout. These are complicated systems and the XRD data (especially in the SI) seems somewhat poor. Also, the STEM-EDS element map signal in Fig 1 seems quite low overall, especially given the dimensions of the nanowires. And, unless I am missing something, the STEM-EDS data in Fig 1 is all that appears to provide somewhat definitive evidence of possible heterostructuring of the materials claimed; the other images are just TEM/HAADF images, which rely on contrast only and therefore could correspond to other materials (Ag instead of Ag_2Te , etc.). Again, unless I am missing something, the materials characterization data are suggestive, but not fully sufficient, to experimentally validate the heterostructuring and its evolution that ultimately is incorporated into the phase field modeling. The authors should also take a closer look at other aspects of the materials characterization in the paper. For example, in Fig. 4, the data is presented in a way that is too small to be useful; I am not able to see most of the characteristic features that are claimed. Also, the XPS data should be fitted. And in Fig S6, the Ag signal (islands) seems too low to be seen.

** Thanks for the valuable comments. Based on the referee’s kind suggestion, we have taken a closer look at the XRD pattern of Te/ Ag_2Te segmented heterostructure. The detailed interpretation is shown as follows based on the XRD pattern of Ag_2Te -Te- Ag_2Te biphasic nanowires previously reported (*Nanoscale*, **2012**, 4, 4537–4543). As shown in Figure, all peaks can be assigned to either hexagonal phase of Te (indexed in blue, PDF#36-1452) or monoclinic phase of Ag_2Te (indexed in

orange, PDF#34-0142). This result confirms the presence of crystalline Te and β - Ag_2Te phases in the same nanostructure. In addition, the lateral dimension of Ag_2Te phase shows an obvious increase in TEM images due to the strain relaxation during the transformation from Te to Ag_2Te . It needs to illustrate that the crystallinity of Te/ Ag_2Te segmented heterostructure is not very good owing to their ultrathin diameter. The other XRD data in SI have also been revised to indicate the heterogeneous structures more clearly in the new version of manuscript.

New Figure S2e. XRD pattern of Te/ Ag_2Te segmented heterostructure.

In addition to the XRD pattern, we also captured the Raman spectra of Te, Ag_2Te and Te/ Ag_2Te NWs, respectively. As shown in Figure, the peaks located at 125 and 2915 cm^{-1} are related to the A_1 mode of Te, corresponding to chain expansion mode in which atom vibrates in the basal plane (*Nano Lett.* **2017**, 17, 3965–3973). The peak located at 650 cm^{-1} is indexed to Ag_2Te , due to its decomposition under laser-beam irradiation (*Opt. Quant. Electron.* **2014**, 46, 573–580). The Raman peaks of Te/ Ag_2Te segmented heterostructures can be ascribed to Te (marked with star) and Ag_2Te (marked with triangle).

New Figure S2f. Raman spectra of Te, Ag_2Te and Te/ Ag_2Te NWs. The wavelength of incident light is 532 nm.

In order to present the segmented heterostructure of Te/ Ag_2Te more clearly, the HADDF image and EDS element mapping have been recaptured. As shown in Figure, the segmented contrast distribution in HADDF image indicates the apparent segmented architectures in the as-prepared

nanowires. Furthermore, Te is continuously distributed throughout the length of nanowire, whereas Ag exists discontinuously in a segmented form. Combined with the result of XRD pattern, the biphasic nature of Te and Ag_2Te in the segmented heterostructure can be proved. And the obvious image drifting in high-resolution mapping is inevitable with the extension of signal acquisition time due to the electron beam drift, electromagnetic interference and other reasons, especially for samples with small sizes (*Nanoscale Adv.*, **2021**, 3, 3035–3040; *Op-to-Electronic Engineering*, **2018**, 45(12), 180198). In addition, the rapid migration of Ag ions under the electron beam significantly affects the signal collection (*J. Am. Chem. Soc.* **2020**, 142(17), 7968–7975).

Figure R16. HADDF image and EDS mapping of Te and Ag in $\text{Te}/\text{Ag}_2\text{Te}$ segmented heterostructure.

Besides, we have also recarried out the mapping collection in the initial island heterostructure. As shown in the following Figure, the islands can be clearly identified. The signal from Te element is uniform throughout the length of NW, while only the areas with higher contrast are distributed with Ag element. Furthermore, to figure out the island composition in the initial heterostructure, we have also retaken the EDS line-scan profiles. The line mapping results in Figure displays the content of Te and Ag as a function of distance, showing that the signal strength of Te is about twice that of Ag in high-contrast areas. This also proves that the islands are Ag_2Te rather than Ag. This is also consistent with the HRTEM image of the island, which shows the lattice fringe of Ag_2Te .

Figure R17. HADDF image, EDS mapping and line mapping of Te and Ag in the initial island heterostructure.

To more clearly represent the structures of these as-synthesized materials, we recollected their representative TEM images at different magnifications. The respective TEM images of G.2 segmented heterostructures ($\text{Te}/\text{Ag}_2\text{Te}$, Te/PbTe , $\text{Te}/\text{Cu}_{1.75}\text{Te}$, $\text{Te}/\text{Bi}_2\text{Te}_3$, Te/CdTe) are shown as bellow. And the corresponding EDS mapping profiles are also presented.

New Figure S1. TEM images of Te/Ag₂Te SHs.

New Figure S15. TEM images of Te/PbTe SHs.

Figure R18. HADF image and element mapping, EDS, and element line mapping of Te/PbTe segmented heterostructures. The scale bar is 20 nm.

New Figure S18. Morphological and structural characterizations of Te/Cu_{1.75}Te SHs.

New Figure S19. Morphological and structural characterizations of Te/CdTe SHs.

New Figure S20. Morphological and structural characterizations of Te/Bi₂Te₃ SHs.

The respective TEM images of the G.3 segmented heterostructures transformed from Te/Ag₂Te (Te/ZnTe, Cu_{1.75}Te/Ag₂Te, CdTe/Ag₂Te, Bi₂Te₃/Ag₂Te, Sb₂Te₃/Ag₂Te, TeSe/Ag₂Te) are shown as below.

New Figure S24. Morphological and structural characterizations of Te/ZnTe SHs.

New Figure S25. Morphological and structural characterizations of $\text{Cu}_{1.75}\text{Te}_3/\text{Ag}_2\text{Te}$ SHs.

New Figure S26. Morphological and structural characterizations of $\text{CdTe}/\text{Ag}_2\text{Te}$ SHs.

New Figure S27. Morphological and structural characterizations of $\text{Bi}_2\text{Te}_3/\text{Ag}_2\text{Te}$ SHs.

New Figure S28. Morphological and structural characterizations of $\text{Sb}_2\text{Te}_3/\text{Ag}_2\text{Te}$ SHs.

The respective TEM images of the G.3 segmented heterostructures transformed from Te/PbTe ($\text{Ag}_2\text{Te}/\text{PbTe}$, $\text{Cu}_{1.75}\text{Te}/\text{PbTe}$, CdTe/PbTe , $\text{Bi}_2\text{Te}_3/\text{PbTe}$, $\text{Sb}_2\text{Te}_3/\text{PbTe}$, Pt/PbTe , Ru/PbTe , Rh/PbTe , Ir/PbTe , $\text{TeSe}/\text{PbTeSe}$) are shown as bellow.

New Figure S31. Morphological and structural characterizations of $\text{Cu}_{1.75}\text{Te}/\text{PbTe}$ SHs.

New Figure S32. Morphological and structural characterizations of CdTe/PbTe SHs.

New Figure S33. Morphological and structural characterizations of Bi₂Te₃/PbTe SHs.

New Figure S34. Morphological and structural characterizations of $\text{Sb}_2\text{Te}_3/\text{PbTe}$ SHs.

New Figure S39. Morphological and structural characterizations of Pt/PbTe SHs.

New Figure S40. Morphological and structural characterizations of Ru/PbTe SHs.

New Figure S41. Morphological and structural characterizations of Rh/PbTe SHs.

New Figure S42. Morphological and structural characterizations of Ir/PbTe SHs.

New Figure S43. Morphological and structural characterizations of TeSe/Ag₂TeSe SHs.

New Figure S44. Morphological and structural characterizations of TeSe/PbTeSe SHs.

Based on the referee's remind, we have recaptured and fitted the XPS data. As shown in Figure, the XPS survey of Te/Ag₂Te segmented heterostructures indicates the presence of both Te and Ag. The Te 3d orbital profile exhibits that Te 3d_{5/2} binding energy of Te/Ag₂Te NWs appears at 572.9 eV, while it is at 572.2 and 572.7 eV for Te and Ag₂Te NWs, respectively. It should be noted that Te NWs can be easily oxidized in the presence of air during measurement. Thus, a pair of oxidation peaks appeared in the 3d region. (Surface characterization at corn of photocathodes for photoinjector applications, doi:10.18429/JACoW-IPAC2015-TUPJE040)

New Figure S2h. Te 3d XPS spectra of Te/Ag₂Te segmented heterostructure.

Besides, based on the referee's remind, we also revised Figure 4 to show the segmented nanostructure library more clearly. Please see the new version of manuscript.

7. Writing: The English/grammar needs to be improved significantly throughout. This is a very complex paper that the authors surely want to be read by researchers across multiple disciplines. Unfortunately, it borders on unreadable at quite a few places. The authors also use incorrect technical terminologies throughout, and should very carefully check this.

** Thanks for the helpful comments. Based on the referee's suggestion, we have tried our best to correct the English throughout the whole text. For example, we have replaced the mechanism

description “near-attracts, far-repels” with “stress induced ordering” to express the driving force of heterostructure formation more accurately. The “vindicated” in “The biphasic attribute is further vindicated by the XRD pattern...” is replaced with “confirmed”. “Due to the high reactivity of Te NWs and the quick reaction with Ag^+ ” is simplified into “Due to the high reactivity of Te NWs with Ag^+ ”. In addition, we have emphatically revised the introduction part to make clear the content of this work. Please see the other changes in the new version of manuscript.

Reviewers' Comments:

Reviewer #3:

Remarks to the Author:

The authors have clearly invested a large amount of time and effort in revising their manuscript including considerable new data and discussion. I believe that they have been very responsive to all of the reviewers and their comments, and have no further suggestions at this stage. While it is true that every manuscript can always be improved, at this stage I believe the work is more than sufficiently complete, and further work - as often considered for any new study - would be better suited to future research and publication(s). I feel the work can now be published and will make a valuable contribution to the field.

Manuscript ID: NCOMMS-23-25379A

“Ordering evolution and post-transformation enabled a library of one-dimensional segmented heteronanostructures”

REVIEWERS' COMMENTS

Reviewer #3 (Remarks to the Author):

The authors have clearly invested a large amount of time and effort in revising their manuscript including considerable new data and discussion. I believe that they have been very responsive to all of the reviewers and their comments, and have no further suggestions at this stage. While it is true that every manuscript can always be improved, at this stage I believe the work is more than sufficiently complete, and further work - as often considered for any new study - would be better suited to future research and publication(s). I feel the work can now be published and will make a valuable contribution to the field.

****We thank the reviewer for strong support on the publication of this work.**